# Learning Materials Interatomic Potentials via Hybrid Invariant-Equivariant Architectures

**Keqiang Yan**[1†*], **Montgomery Bohde**[1*] , **Andrii Kryvenko**[1*] , **Ziyu Xiang**[1], **Kaiji Zhao**[1], **Siya Zhu**[1], **Saagar Kolachina**[1], **Doğuhan Sarıtürk**[1], **Jianwen Xie**[2], **Raymundo Arróyave**[1], **Xiaoning Qian**[1,3], **Xiaofeng Qian**[1], **Shuiwang Ji**[1†]

[1]*Texas A&M University*   [2]*Lambda Inc.*   [3]*Brookhaven National Laboratory*

Reviewed on OpenReview: *https://openreview.net/forum?id=fq3nrVqNmL*

## Abstract

Machine learning interatomic potentials (MLIPs) can predict energy, force, and stress of materials and enable a wide range of downstream discovery tasks. A key design choice in MLIPs involves the trade-off between invariant and equivariant architectures. Invariant models offer computational efficiency but may not perform as well, especially when predicting high-order outputs. In contrast, equivariant models can capture high-order symmetries, but are computationally expensive. In this work, we propose HIENet, a hybrid invariant-equivariant materials interatomic potential model that integrates both invariant and equivariant message passing layers. We show that HIENet provably satisfies key physical constraints and achieves strong empirical performance across a wide range of benchmarks and downstream discovery tasks. Further evaluations show that HIENet achieves considerable computational speedups compared to prior equivariant models. Finally, additional ablations demonstrate that our hybrid invariant-equivariant approach scales well across model sizes and works with different equivariant model architectures, providing insights into future MLIP designs.

## 1 Introduction

The discovery of materials with desired properties underpins a wide range of technological advancements (de Pablo et al., 2019; Stach et al., 2021; Shafian et al., 2025; Lv et al., 2022; Zheng et al., 2021; Miracle & Thoma, 2024). However, traditional materials discovery relies heavily on costly trial-and-error experimental methods. Computational approaches, particularly those leveraging advanced quantum mechanical methods such as density functional theory (DFT), have accelerated this process (Zhang et al., 2023), but despite their benefits, simulating systems with a large number of atoms remains extremely expensive.

Recent progress in machine learning interatomic potentials (MLIPs) offers a promising path forward by enabling the prediction of energies, forces, and stresses of materials while achieving significant speedups compared to traditional DFT methods. However, existing MLIP models still face a fundamental trade-off: invariant models are computationally efficient but struggle with high-order property predictions, while equivariant models can better capture high-order interactions but are computationally expensive.

An additional design choice is whether to enforce model predictions to adhere to key physical constraints detailed in Sec. 3.1. Recent works have tried to learn these physical constraints, such as EquiformerV2 (Liao et al., 2024), which enforces global symmetry operations but not the other physical laws, and ORB (Neumann et al., 2024), which doesn't impose any constraints on model predictions. While it is more computationally expensive to enforce these physical constraints, it is also necessary for MLIPs to perform well, especially on downstream discovery tasks beyond energy, force, and stress prediction.

In this work, we propose HIENet, a materials MLIP that satisfies key physical constraints for energy, force, and stress predictions while integrating both invariant and equivariant designs to achieve strong performance

---

*Equal contribution. †Corresponding authors.

with considerable computational speedups compared to existing models. An overview of HIENet is provided in Figure 1. Unlike prior approaches that rely exclusively on either invariant or equivariant layers, HIENet balances these strategies to leverage the scalability of invariant layers while utilizing equivariant layers to effectively capture high-order interactions. Moreover, in contrast to existing models like EquiformerV2 (Liao et al., 2024), which enforces O(3) equivariance but violates force conservation through direct force prediction, and ORB (Neumann et al., 2024), HIENet rigorously satisfies physical constraints, including O(3) equivariance for force and stress, and adheres to physical conservation laws through physics-informed derivative-based methods. Experimental results on common benchmarks including Materials Project Trajectory, Matbench Discovery, and downstream materials discovery tasks including evaluations on phonons, bulk moduli, *ab initio* molecular dynamics, and alloys as detailed in Sec. 5.1, 5.2, 5.3 and Appendix B,C demonstrate the efficiency and effectiveness of HIENet. Additional ablations in Sec. 5.5 further demonstrate the generality of our hybrid invariant-equivariant approach across different model capacities and equivariant layer designs.

## 2 Related Work

In this section, we focus on materials MLIPs and provide related works on conventional computation methods in Appendix A.2. Recent advances in materials property prediction models (Xie & Grossman, 2018; Choudhary & DeCost, 2021; Yan et al., 2022; Lin et al., 2023; Choudhary et al., 2024; Yan et al., 2024) and the availability of high-quality materials dynamics datasets (Chen & Ong, 2022; Deng et al., 2023a; Barroso-Luque et al., 2024) generated using DFT-based algorithms have facilitated the development of powerful materials MLIPs. Among these MLIPs, models with only invariant layers, such as M3GNet (Chen & Ong, 2022), CHGNet (Deng et al., 2023a), ORB (Neumann et al., 2024), and EScAIP (Qu & Krishnapriyan, 2024), are computationally efficient but struggle to produce physically meaningful and robust predictions, especially on downstream tasks. In contrast, models with purely equivariant layers, including MACE (Batatia et al., 2023), SevenNet (Park et al., 2024a), and EquiformerV2 (Barroso-Luque et al., 2024), are more powerful, but also more computationally expensive. Their extensive use of tensor product operations limits their scalability. Moreover, even some equivariant models, such as EquiformerV2, violate force conservation—either through direct force prediction or discontinuities in the predicted potential energy surface—undermining their utility in realistic materials tasks as seen in Sec. 5. Several more recent MLIPs have been proposed that achieve strong performance on relevant benchmarks, including purely invariant models such as MatRIS (Zhou et al.) and DPA-3.1 (Zhang et al., 2025), equivariant models such as eSEN (Fu et al., 2025), GRACE-2L (Lysogorskiy et al., 2026), and Nequix (Koker et al., 2025), and hybrid designs such as Eqnorm (Chen, 2025). A detailed comparison of these models is provided in Appendix E.

Different from these existing MLIPs, our proposed HIENet satisfies the key physical constraints outlined in Sec. 3.1 and combines the scalability and efficiency of invariant designs with the robustness and symmetry-capturing capabilities of equivariant designs. This novel integration offers a promising direction for the next generation of MLIPs design.

## 3 Preliminaries

**Problem definition**. The core task in developing MLIPs is to learn a mapping from materials atomic structures to quantum mechanical properties. Specifically, given a crystal structure, we aim to predict three quantities; the total energy $E$, the forces acting on each atom $\mathbf{F} = \{\mathbf{F}_i \in \mathbb{R}^3, 1 \le i \le n\}$, where $n$ denotes the number of atoms in a cell, and the stress tensor $\boldsymbol{\sigma} \in \mathbb{R}^{3 \times 3}$, which governs cell deformation. While these properties are directly useful for many applications such as structural relaxation and predicting thermodynamic stability, from these we can derive many other important material properties such as phonon band structures and bulk moduli as shown in Sec. 5.3. We also provide preliminaries about molecular dynamics simulation of materials in Appendix A.

**Crystal structures**. Unlike regular molecules, crystals are periodic in nature and are characterized as three-dimensional lattices with infinitely repeating unit cells. Adopting the notation of Yan et al. (2024), a crystal structure can be described as a triple $\mathbf{M} = (\mathbf{Z}, \mathbf{P}, \mathbf{L})$, which represents both atomic and geometric information. The atomic composition is denoted by $\mathbf{Z} = [z_1, z_2, \cdots, z_n] \in \mathbb{Z}^n$, where $z_i$ represents the atomic

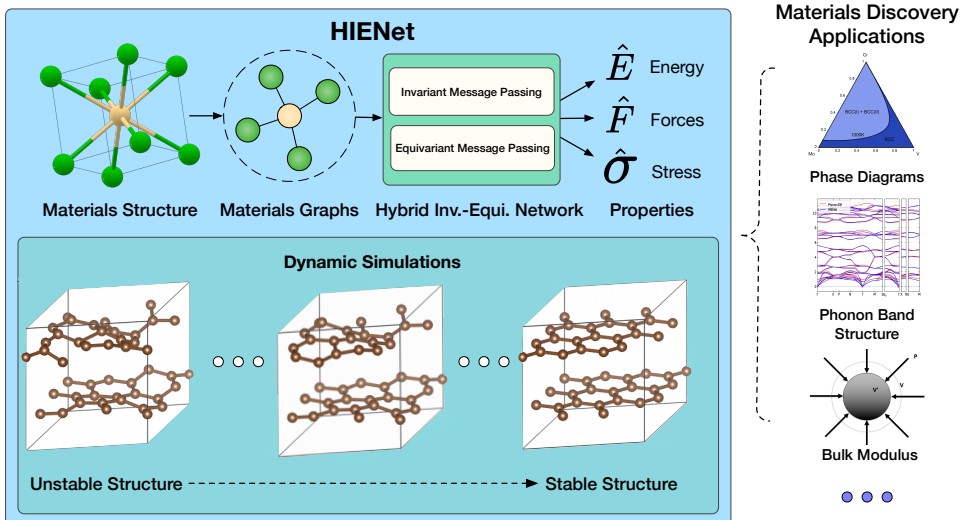

Figure 1: **HIENet overview**. The model converts material structures into graph representations and processes them through a hybrid architecture combining invariant and equivariant message passing networks to predict physical properties. The model supports accurate dynamic simulations (bottom) and enables diverse materials science applications (right).

number of $i$-th atom in the unit cell. The arrangement of these atoms in Euclidean space is given by 3D coordinates $\mathbf{P} = [\mathbf{p}_1, \mathbf{p}_2, \cdots, \mathbf{p}_n] \in \mathbb{R}^{3 \times n}$. The periodicity of the unit cell is specified by the lattice matrix $\mathbf{L} = [\boldsymbol{\ell}_1, \boldsymbol{\ell}_2, \boldsymbol{\ell}_3] \in \mathbb{R}^{3 \times 3}$, whose columns are the three lattice vectors.

## 3.1 Physical Constraints for MLIPs

As stated previously, MLIPs are developed to predict energy, forces, and stress of materials atomic systems. While some MLIPs such as EqV2 and ORB can achieve moderate performance on some tasks without satisfying key physical constraints, such models under-perform on many important materials discovery tasks such as phonon frequency calculations, bulk moduli prediction, and molecular dynamics simulations, as demonstrated empirically in Sec. 5.3. As such, in order for MLIPs to generalize well and have robust performance across downstream tasks, it is essential that model predictions satisfy key physical constraints.

**Rototranslational Symmetries**. Crystal structures exhibit inherent symmetry under global rotations, translations, and reflections. To respect these symmetries, the predicted energy must be E(3) invariant, while forces and stress must be O(3) equivariant. We formalize these requirements as follows:

**Definition 3.1** (O(3) Equivariance). An MLIP produces *O(3) equivariant predictions* if, for a crystal structure $\mathbf{M} = (\mathbf{Z}, \mathbf{P}, \mathbf{L})$, its predicted energy $\hat{E}$, forces $\hat{\boldsymbol{F}} = (\hat{\boldsymbol{F}}_1, \ldots, \hat{\boldsymbol{F}}_n)$, and stress tensor $\hat{\boldsymbol{\sigma}}$ transform under any rotation $\mathbf{R} \in \mathbb{R}^{3 \times 3}$, $|\mathbf{R}| = \pm 1$ and translation $\mathbf{b} \in \mathbb{R}^3$ as follows:

$$\hat{E}(\mathbf{Z}, \mathbf{P}, \mathbf{L}) = \hat{E}(\mathbf{Z}, \mathbf{RP} + \mathbf{b}, \mathbf{RL})$$
$$\hat{\boldsymbol{F}}_i(\mathbf{Z}, \mathbf{P}, \mathbf{L}) = \mathbf{R}^\top \hat{\boldsymbol{F}}_i(\mathbf{Z}, \mathbf{RP} + \mathbf{b}, \mathbf{RL}),$$
$$\hat{\boldsymbol{\sigma}}(\mathbf{Z}, \mathbf{P}, \mathbf{L}) = \mathbf{R}^\top \hat{\boldsymbol{\sigma}}(\mathbf{Z}, \mathbf{RP} + \mathbf{b}, \mathbf{RL})\mathbf{R}.$$

Importantly, there is a distinction between invariant/equivariant layers and invariant/equivariant predictions. For example, CHGNet (Deng et al., 2023b) exclusively uses E(3) invariant message passing layers, yet CHGNet force and stress predictions are O(3) equivariant because they use gradient-based force and stress calculations. When we refer to a model being O(3) equivariant, we are referring to the outputs, not the individual layers, unless otherwise specified.

**Physical Plausibility**. Beyond symmetry considerations, MLIPs must satisfy several key physical laws to be reliable for downstream applications. These include force conservation, force equilibrium, and stress tensor symmetry. We define these constraints formally as follows:

**Definition 3.2** (Force Conservation and Equilibrium). Forces must form a conservative vector field derived from the potential energy surface, and in the absence of external influences, the sum of forces on all atoms is zero:

$$\mathbf{F} = -\nabla_{\mathbf{P}} E, \quad \sum_{i=1}^{n} \mathbf{F}_i = \mathbf{0}, \tag{1}$$

**Definition 3.3** (Stress Tensor Symmetry). The predicted stress tensor must be symmetric:

$$\sigma_{ij} = \sigma_{ji} \quad \forall i, j \in 1, 2, 3 \tag{2}$$

In addition to these physical laws, the potential energy surface must be continuously differentiable to enable accurate downstream property calculations requiring higher-order derivatives.

Our HIENet model rigorously enforces all the outlined symmetry and physical constraints through the carefully designed geometric crystal graphs, model architecture, and gradient-based force and stress computation, with details provided in Sec. 4.1, 4.2, 4.3, and 4.4.

## 4 Hybrid Invariant-Equivariant Networks

We propose Hybrid Invariant-Equivariant Network (HIENet), a materials interatomic potential model that integrates both invariant and equivariant message passing layers. HIENet is carefully designed to satisfy important physical constraints detailed in Sec. 3.1, consisting of physics-informed geometric crystal graphs detailed in Sec. 4.1, a hybrid invariant-equivariant network design detailed in Sec. 4.2, and physics-informed property predictions detailed in Sec. 4.3. All together, HIENet achieves improved performance on common benchmarks and downstream materials discovery tasks while significantly improving computational efficiency compared to prior models, as evaluated against EquiformerV2 (Liao et al., 2024), SevenNet (Park et al., 2024b), MACE (Batatia et al., 2023), ORB (Neumann et al., 2024), and CHGNet (Deng et al., 2023b) in Sec. 5. Additionally, HIENet satisfies the desirable physical constraints from Sec. 3.1 with mathematical proofs in Sec. 4.4.

### 4.1 Geometric Graph Representations Satisfying Physical Constraints

For a crystal structure, $\mathbf{M} = (\mathbf{Z}, \mathbf{P}, \mathbf{L})$ we construct an O(3) equivariant crystal graph $G = (V, E)$ that preserves physical symmetries inherent in crystal structures. Specifically, for a crystal structure $\mathbf{M} = (\mathbf{Z}, \mathbf{P}, \mathbf{L})$, each atom $i$ in the unit cell and all its periodic duplicates are represented by a single node $i \in V$ with node features $\mathbf{h}_i = \mathbf{W}_{\text{emb}} \mathbf{z}_i$, where $\mathbf{W}_{\text{emb}} \in \mathbb{R}^{d \times n_z}$ is a learnable embedding matrix and $\mathbf{z}_i$ is the one-hot encoding of atomic number $z_i$. An edge will be built from node $j$ to $i$ when the Euclidean distance between a periodic duplicate $j'$ of $j$ and $i$ satisfies

$$||\boldsymbol{r}_{j'i}||_2 = ||\mathbf{p}_j + k_1\ell_1 + k_2\ell_2 + k_3\ell_3 - \mathbf{p}_i||_2 \leq R_{\text{cut}}, \ k_1, k_2, k_3 \in \mathbb{Z}, \tag{3}$$

where $R_{\text{cut}}$ is a fixed cutoff radius.

Edge features $\mathbf{h}_{ji}$ are then embedded using radial Bessel basis functions with a polynomial envelope (Gasteiger et al., 2021) function $f_{\text{poly}}$:

$$\boldsymbol{h}_{ji} = \frac{2\sin\left(\frac{n\pi}{R_{\text{cut}}}||\boldsymbol{r}_{ji}||_2\right)}{R_{\text{cut}}||\boldsymbol{r}_{ji}||_2} f_{\text{poly}}\left(||\boldsymbol{r}_{ji}||_2, R_{\text{cut}}\right). \tag{4}$$

**Importance of using envelope function**. It is worth noting that the smooth envelope is crucial for energy conservation and computing physically meaningful force predictions. It ensures that the energy and its derivatives smoothly decay to zero at the cutoff boundary.

**Importance of O(3) equivariant crystal graphs**. While radius-based graph construction is standard, some prior crystal graph works (Yan et al., 2022; 2024) additionally incorporate SO(3) equivariant periodic encodings to achieve geometric completeness, which however breaks O(3) equivariance. In contrast, our crystal graphs are O(3) equivariant by construction, as the use of a fixed cutoff radius and edge vectors $\boldsymbol{r}_{ji}$

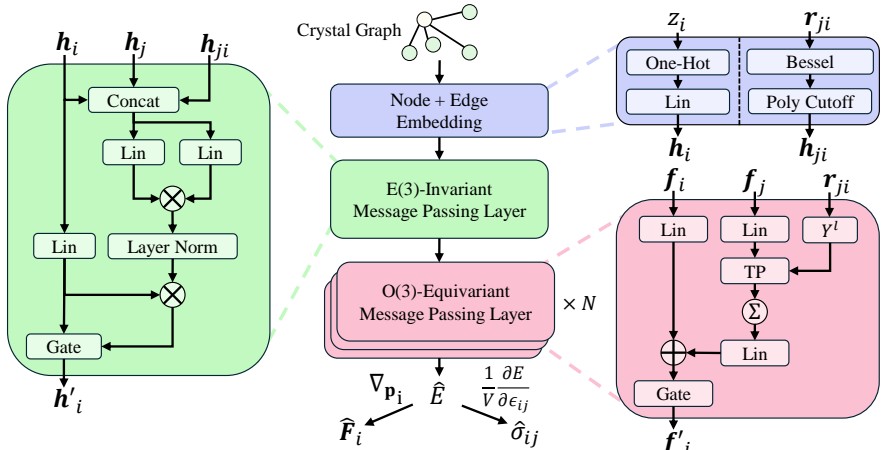

Figure 2: **HIENet Model Architecture**. We construct O(3) equivariant crystal graphs. We then apply an invariant message passing layer followed by several equivariant message passing layers before predicting the total energy, $\hat{E}$ and using physical laws to compute $\hat{F}$, $\hat{\sigma}$.

are O(3) equivariant by definition. It is important to note that O(3) equivariance of the input crystal graphs is a necessary condition for achieving O(3) equivariant predictions in MLIPs. Moreover, since the underlying DFT calculations are inherently O(3) equivariant, such SO(3) equivariant encodings introduce a symmetry mismatch that leads to a measurable drop in MLIP performance, as shown in Sec. 5.6.

## 4.2 Hybrid Invariant-Equivariant Design

Our HIENet model consists of several invariant and equivariant message passing layers which iteratively update node features for each atom.

**E(3) Invariant Layer.** In the invariant message passing layers, we update node features $h_i$ using a graph transformer mechanism. Specifically, we compute key $k_{ji}$, query $q_{ji}$, and value $v_{ji}$ vectors as:

$$k_{ji} = \mathbf{W}_K\left(h_i||h_j||h_{ij}\right), \quad q_{ji} = \mathbf{W}_Q\left(h_i||h_j||h_{ij}\right), \quad v_{ji} = \Phi\left(h_i||h_j||h_{ji}\right), \tag{5}$$

where $\Phi$ is an MLP with SiLU nonlinearities (Elfwing et al., 2018) and $||$ denotes vector concatenation. We then compute attention scores and aggregate the values over neighboring nodes to update the node features:

$$h_i' = \varphi(h_i) + (1 - \varphi(h_i)) \sum_{j \in \mathcal{N}_i} v_{ji} \odot \sigma\left(\frac{q_{ji} \odot k_{ji}}{\sqrt{d}}\right), \tag{6}$$

where $\odot$ represents element-wise multiplication, $\sigma$ is the sigmoid activation function, and $\varphi$ is an MLP with a sigmoid activation in the final layer that acts as a learnable gating mechanism.

**O(3) Equivariant Layer.** The HIENet equivariant layers build node features $f_{i,\ell}$ for each rotation order $\ell \leq L_{\max}$. In practice, we use $L_{\max} = 3$. In the first equivariant message passing layer, we initialize $f_{i,0} = h_i$, the output of the previous invariant layer. In each equivariant layer, we embed edge vectors $r_{ji}$ using spherical harmonics $Y^l(\frac{r_{ji}}{||r_{ji}||})$ and update the equivariant features as:

$$f_{i,\ell}' = \frac{1}{|\mathcal{N}_i|} \sum_{l=0}^{L_{\max}} \sum_{j \in \mathcal{N}_i} \mathbf{TP}_\ell\left(\mathbf{W}f_i, Y^l\left(\frac{r_{ji}}{||r_{ji}||}\right)\right) \tag{7}$$

where $\mathbf{TP}_\ell$ is the standard Clebsch-Gordan tensor product operation yielding outputs with rotation order $\ell$, and $\mathbf{W}f_i = \bigoplus_{\ell=0}^{L_{\max}} \mathbf{W}_\ell f_{i,\ell}$ represents per-order linear mixing with learnable weight matrices $\mathbf{W}_\ell$ applied independently to features of each rotation order $\ell$. We further add a skip connection and gate mechanism to output updated node features:

$$f_i' = \psi\left(\mathbf{W}_{\text{skip}}f_i + \mathbf{W}_E f_i'\right) \tag{8}$$

where $\psi$ is an equivariant gate activation function defined as:

$$\psi(\boldsymbol{f}_i) = \phi(\boldsymbol{f}_{i,0}) \oplus \left( \bigoplus_{0 < l \leq L_{\max}} \phi(\boldsymbol{f}_{i,0}) \boldsymbol{f}_{i,l} \right) \tag{9}$$

where $\phi$ is the SiLU activation function and $\oplus$ is the direct sum operation.

**Hybrid MLIPs**. Our HIENet model consists of one or more invariant message passing layers followed by several equivariant layers. The final equivariant layer outputs are aggregated across the graph and the energy is predicted as $E = \sum_i \mathbf{W}_e \boldsymbol{f}_{i,0}$. In practice, we find that one invariant message passing layer achieves a good balance between performance and efficiency. We provide additional details and ablations on the model design in Sec. 5.6, as well as Appendix F and G.

### 4.3 Physics Informed Property Predictions

In order to ensure that our force and stress predictions obey the aforementioned physical constraints, we use gradient-based methods to compute force and stress. Specifically, our model directly predicts the total energy, $\hat{E}$ and we compute the force acting on atom $i$ as $\hat{\boldsymbol{F}}_i = -\nabla_{\boldsymbol{p}_i}\hat{E}$, where $\nabla_{\boldsymbol{p}_i}$ represents the gradient with respect to the position vector $\boldsymbol{p}_i$. This approach automatically guarantees that:

**Proposition 4.1.** *HIENet predictions $\hat{\boldsymbol{F}}_i$ form a conservative vector field.*

**Proposition 4.2.** *HIENet predictions satisfy force equilibrium $\sum_{i=1}^{N} \hat{\boldsymbol{F}}_i = \mathbf{0}$ when no external influences are applied.*

Similarly, we compute the stress tensor through strain derivatives $\hat{\sigma}_{ij} = \frac{1}{V}\frac{\partial \hat{E}}{\partial \epsilon_{ij}}$, where $\boldsymbol{\epsilon}$ is the lattice strain tensor and $V$ is the volume of the unit cell. We ensure that $\hat{\boldsymbol{\sigma}}$ will be symmetric by first symmetrizing the strain matrix $\boldsymbol{\epsilon}_{\text{sym}} = \frac{1}{2}(\boldsymbol{\epsilon} + \boldsymbol{\epsilon}^\top)$. All together, our approach guarantees that:

**Proposition 4.3.** *HIENet predictions are O(3) equivariant as defined in Sec. 3.1.*

### 4.4 Proofs of Satisfying Physical Constraints

While previous works (Schütt et al., 2017; 2021; Chen & Ong, 2022; Deng et al., 2023b; Park et al., 2024b) used gradient-based calculations, none of these works prove that their proposed methods satisfy desired physical laws. In this section, we rigorously prove each of the previously stated propositions and show that HIENet satisfies the desirable physical constraints.

*Proof of Proposition 4.1.* By definition, a vector field $\mathbf{v} : \mathbb{R} \to \mathbb{R}^n$ is conservative if there exists a continuously differentiable scalar field $\varphi$ such that $\mathbf{v} = \nabla \varphi$.

We compute forces as $\hat{\boldsymbol{F}} = -\nabla_{\mathbf{P}}\hat{E}$, so HIENet predictions form a conservative force-field as long as the energy $E$ is continuously differentiable with respect to the atom positions, $\mathbf{p}_i$. Clearly, each of the operations in HIENet, linear transforms, SiLU (Elfwing et al., 2018) activations, spherical harmonics $Y^l(\frac{r_{ji}}{||r_{ji}||})$, and edge embedding functions, are continuously differentiable within the domain of possible interatomic distances, i.e. for $||r_{ji}|| \neq 0$. Importantly, $f_{\text{poly}}$ is specifically chosen so that it is continuously differentiable and decays to 0 at $R_{\text{cut}}$ (Gasteiger et al., 2021). Because each operation in HIENet is continuously differentiable, the force predictions therefore form a conservative vector field, as desired. $\square$

*Proof of Proposition 4.2.* We define edge force as $\hat{\boldsymbol{F}}_{ji} = -\frac{\partial \hat{E}}{\partial \boldsymbol{r}_{ji}}$.

Forces acting on each atom can be decomposed as:

$$\hat{\boldsymbol{F}}_i = -\frac{\partial \hat{E}}{\partial \boldsymbol{p}_i} = -\sum_{j \in \mathcal{N}_i} \left( \frac{\partial \hat{E}}{\partial \boldsymbol{r}_{ji}} \frac{\partial \boldsymbol{r}_{ji}}{\partial \boldsymbol{p}_i} + \frac{\partial \hat{E}}{\partial \boldsymbol{r}_{ij}} \frac{\partial \boldsymbol{r}_{ij}}{\partial \boldsymbol{p}_i} \right) \tag{10}$$

$$= -\sum_{j \in \mathcal{N}_i} \left( \frac{\partial \hat{E}}{\partial \boldsymbol{r}_{ji}} - \frac{\partial \hat{E}}{\partial \boldsymbol{r}_{ij}} \right) = \sum_{j \in \mathcal{N}_i} \left( \hat{\boldsymbol{F}}_{ji} - \hat{\boldsymbol{F}}_{ij} \right) \tag{11}$$

Because the graph construction is symmetric, i.e. $\boldsymbol{r}_{ji} = \boldsymbol{r}_{ij}$:

$$\sum_{i=1}^{n} \hat{\boldsymbol{F}}_i = \sum_{i=1}^{n} \sum_{j \in \mathcal{N}_i} \left( \hat{\boldsymbol{F}}_{ji} - \hat{\boldsymbol{F}}_{ij} \right) = \sum_{(i,j) \in \mathcal{E}} \left( \hat{\boldsymbol{F}}_{ji} - \hat{\boldsymbol{F}}_{ij} \right) = 0 \tag{12}$$

Therefore, the forces acting on each atom sum to 0 as desired. $\qquad\square$

*Proof of Proposition 4.3.* We provide a sketch of the proof idea with additional details in Appendix H.

HIENet energy predictions are E(3) invariant: the invariant layers are E(3) invariant by construction, and for the equivariant layers, we only extract the final $l = 0$ features, which are invariant. Because the energy predictions are invariant and because we use gradient-based property predictions described in Sec. 4.3, the force, and stress predictions will be O(3) equivariant as desired. $\qquad\square$

An important point, also noted by Fu et al. (2025), is that the graph-construction method can prevent the model from satisfying key physical constraints. For example, previous works such as Yan et al. (2022); Liao et al. (2024) use nearest-neighbor graph construction, which is not continuously differentiable. In Yan et al. (2024) additional edge vectors are added which will violate the condition that $\hat{\boldsymbol{F}}_{ji} = \hat{\boldsymbol{F}}_{ij}$ in the proof of Proposition 4.2, causing the forces to not sum to 0, and the model predictions to not form a conservative vector field.

## 5 Experimental Evaluations

In this section, we evaluate HIENet's overall modeling capacity as an MLIP. We assess its performance on the widely used Matbench Discovery benchmark (Riebesell et al., 2023) and Materials Project Trajectory (MPtrj) dataset (Deng et al., 2023a) in Sec. 5.1 and 5.2. We then provide evaluations on important downstream materials discovery tasks in Sec. 5.3, where we find that models which do not satisfy physical constraints perform poorly. In Sec. 5.4 we evaluate computational efficiency and demonstrate that HIENet is able to achieve superior performance across all tasks while still providing considerable computational speedups. Finally, in Sec. 5.5 we provide ablations studies to demonstrate the robustness and generality of our hybrid invariant-equivariant network design. We provide additional downstream materials discovery evaluations on *ab initio* molecule dynamics simulations and phase diagram prediction for alloy design in Appendix B and Appendix C.

**Experimental setup**. We train our HIENet on the MPtrj dataset (Deng et al., 2023b), which contains 1.58M crystal structures. We split the dataset and use 95% for training and 5% for validation following Batatia et al. (2023). For fair comparison, we compare with models trained on this dataset and without any auxiliary data or training objectives. Specifically, we compare against state-of-the-art methods including EquiformerV2 (Liao et al., 2024), ORB (Neumann et al., 2024), SevenNet (Park et al., 2024b), MACE (Batatia et al., 2023), and CHGNet (Deng et al., 2023b). Of these baselines, all but ORB have equivariant force and stress predictions, and all but EquiformerV2 and ORB satisfy the physical constraints listed in Sec. 3.1. More detailed model settings and training details can be found in Appendix G. In all tables, we mark best performing model in **bold** and second best in underlined.

### 5.1 Evaluation on Matbench Discovery

Matbench Discovery benchmark (Riebesell et al., 2023) is a comprehensive testbed for crystalline materials structure optimizations and stability predictions. Notably, the Matbench Discovery benchmark structures come from a different distribution from the MPtrj training dataset, thus posing an out-of-distribution (OOD) generalization problem. As shown in Table 1, HIENet performs best on **all seven** metrics and has a significant performance gain on the Discovery Acceleration Factor (DAF). Additionally, we observe that while ORB performs well on all of the energy-related metrics, it has the worst performance of all models on RMSD, a metric that measures the models ability to accurately relax structures to stability. This aligns with our intuition that downstream tasks such as structural optimization require models to obey physical symmetries.

Table 1: Model performance on the Unique Prototype split of the Matbench Discovery benchmark. DAF is the Discovery Acceleration Factor from Riebesell et al. (2023) which measures model performance to classify thermodynamic stability. MAE and RMSE are in meV/atom. RMSD is the root mean squared displacement between predicted and reference structures after relaxation. Missing results from corresponding model marked by -. Comparisons with more recent models are provided in Appendix E.

| Model | HIENet | EquiformerV2 | ORB | SevenNet-l3i5 | MACE | CHGNet |
|---|---|---|---|---|---|---|
| DAF ↑ | **4.93** | 4.64 | 4.70 | 4.63 | 3.78 | 3.361 |
| MAE ↓ | **41** | 42 | 45 | 48 | 57 | 63 |
| RMSE ↓ | **84** | 87 | 91 | 87 | 101 | 103 |
| $R^2$ ↑ | **0.793** | 0.778 | 0.756 | 0.776 | 0.697 | 0.689 |
| F1 ↑ | **0.777** | 0.77 | 0.765 | 0.76 | 0.669 | 0.613 |
| Accuracy ↑ | **0.93** | **0.93** | 0.92 | 0.92 | 0.88 | 0.85 |
| Precision ↑ | **0.754** | 0.709 | 0.719 | 0.708 | 0.577 | 0.514 |
| RMSD ↓ | **0.080** | - | 0.101 | 0.085 | 0.091 | 0.095 |

Table 3: Error in phonon frequency prediction on structures from Riebesell & Naik (2024). MAE and MSE computed against each q-point, and RMSE taken as the root of MSE over all q-points and bands. Band structures shown in Appendix D.1

| Model | HIENet | EquiformerV2 | ORB | SevenNet-l3i5 | MACE | CHGNet |
|---|---|---|---|---|---|---|
| MAE (THz) | **0.316** | 1.359 | 1.601 | 0.325 | 0.529 | 1.359 |
| MSE (THz$^2$) | **0.332** | 4.65 | 5.441 | 0.358 | 0.837 | 4.21 |
| RMSE (THz) | **0.446** | 1.657 | 1.973 | 0.455 | 0.699 | 1.604 |

## 5.2 Materials Project Trajectory Dataset

We then evaluate HIENet's ability to accurately predict energy, force, and stress on a held-out MPtrj validation set following Deng et al. (2023a); Batatia et al. (2023). SevenNet does not hold-out any validation split and trains their model on the entire 1.58M structures. To compare with SevenNet, we also report HIENet performance on the training split. As seen in Table 2, HIENet outperforms all baseline methods across train and validation splits. Notably, HIENet reduces the energy mean absolute error (MAE) by nearly 50% and the force MAE by 23% compared to the next best method, EquiformerV2.

Table 2: MAE on train and validation splits. Inv. and Eqv. denote whether the model uses invariant or equivariant message passing layers. ORB (Neumann et al., 2024) does not report results on MPtrj and MACE does not report stress performance.

| Model | Inv. | Eqv. | Energy ↓ (meV/atom) | Forces ↓ (meV/Å) | Stress ↓ (kBar) |
|---|---|---|---|---|---|
| Train | | | | | |
| SevenNet-0 | ✗ | ✓ | 11.5 | 41 | 2.78 |
| SevenNet-l3i5 | ✗ | ✓ | 8.3 | 29 | 2.33 |
| HIENet | ✓ | ✓ | **5.91** | **20.76** | **1.95** |
| Validation | | | | | |
| CHGNet | ✓ | ✗ | 33 | 79 | 3.51 |
| MACE | ✗ | ✓ | 20 | 45 | - |
| EquiformerV2 | ✗ | ✓ | 12.4 | 32.22 | 2.48 |
| HIENet | ✓ | ✓ | **6.77** | **24.82** | **2.31** |

## 5.3 Evaluations on Phonons and Bulk Modulus Prediction

**Phonon frequency evaluation**. Phonons are collective excitations of atomic vibrations in crystal structures with translational symmetry, playing a crucial role in determining the dynamical stability and thermal conductivity of materials. The calculation of phonon band structure relies on the atomic forces upon displacement of atoms in different phonon modes along high-symmetry paths in the first Brillouin zone. Because of this, it is critical for model predictions to obey physical symmetries and for the forces to be conservative. We perform a phonon band structure calculation workflow using Phonopy (Togo et al., 2023;

Table 4: Error in bulk modulus $K_{VRH}$ prediction across 1,763 crystal structures sampled from Material Project.

| Model | HIENet | EquiformerV2 | ORB | SevenNet-l3i5 | MACE | CHGNet |
|---|---|---|---|---|---|---|
| MAE (GPa) ↓ | **10.52** | 24.76 | 34.7 | 11.5 | 28.84 | 21.67 |
| $R^2$ ↑ | **0.93** | 0.64 | -21.9 | 0.9 | -54.1 | 0.7 |

Togo, 2023) on the set of structures from Riebesell & Naik (2024). Additional results and details of our workflow are provided in Appendix D.1.

As shown in Table 3, HIENet outperforms all baseline models across all metrics for phonon frequency calculations. Additionally, while EquiformerV2 and ORB have good performance on MPtrj and Matbench Discovery, they have the worst performance among all models on this task. As previously mentioned, this may be because this task requires models predictions to obey physical constraints and be physically meaningful in order for the phonon calculations to be accurate.

**Evaluation on bulk modulus**. Model efficacy on zero-shot prediction of material properties was further evaluated on calculations of the fourth-order elastic tensor and the corresponding VRH average bulk modulus $K_{VRH}$ (Hill, 1952). A test set was generated by querying the Materials Project Database (Jain et al., 2013) for entries with between 1 and 6 sites that also had reported elasticity values. Following Batatia et al. (2023), we remove entries with highly unphysical bulk modulus reference values less than -50 GPa or greater than 600 GPa as well as those resulting in a calculated singular matrix, resulting in a final evaluation set of 1,763 crystal systems. Elastic tensors and bulk moduli were computed using the MatCalc's Elasticity module (Liu et al., 2024). Additional details on our workflow and dataset are provided in Appendix D.2.

As shown in Table 4, HIENet outperforms all models on MAE and $R^2$. In fact, HIENet and SevenNet are the only models capable of achieving reasonable accuracy, demonstrating both the difficulty of this task and the robustness of our model. We provide parity plots for all models in Appendix D.2.

## 5.4 HIENet efficiency

In addition to demonstrating improved performance across all benchmarks and downstream tasks, we show that HIENet is more computationally efficient than competing equivariant models. This is highly important for downstream materials discovery applications such as structural relaxation and random structure search, which require thousands of forward passes of the model. As seen in Table 5, HIENet is 90% faster than SevenNet-l3i5 and over 140% faster than EquiformerV2, all while having better performance than both models. Both EquiformerV2 and SevenNet exclusively use equivariant message pass-

Table 5: Number of parameters and inference throughput of HIENet compared with top performing equivariant models. Throughput evaluated using random samples from the MPtrj dataset on a single Nvidia A100 GPU with batch size 1.

| Model | Num. of Param. | Throughput ↑ (Samples / sec.) |
|---|---|---|
| SevenNet-l3i5 | 1,171,327 | 11.9 |
| EquiformerV2 | 31,207,434 | 9.4 |
| HIENet | 7,510,280 | 22.6 |

ing layers, which limits the throughput and scalability of these models. At the same time, models without O(3) equivariant force and stress predictions, such as ORB, may be faster, but will perform poorly on realistic materials discovery tasks, as shown in Sec. 5.1 and 5.3.

## 5.5 Generality and Robustness of Hybrid Network Design

**Hybrid architecture ablation**. We provide an ablation study to demonstrate that our hybrid invariant-equivariant architecture outperforms invariant-only and equivariant-only models. Specifically, we construct increasingly large models by varying the model width (the dimension of the invariant/equivariant features). We fix the number of layers and all other model design choices. In Figure 3 we see that HIENet outperforms EqvNet (only equivariant layers) and InvNet (only invariant layers) across a range of model sizes. Additionally,

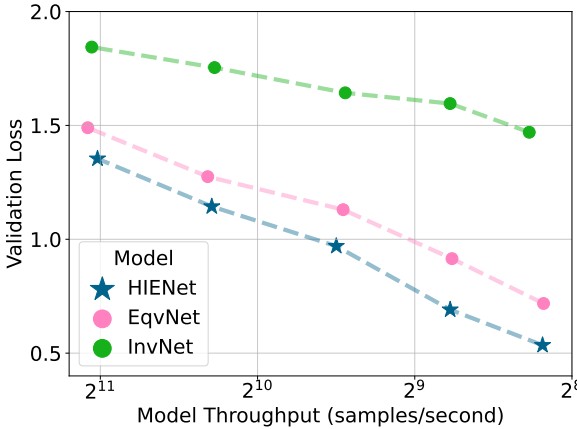 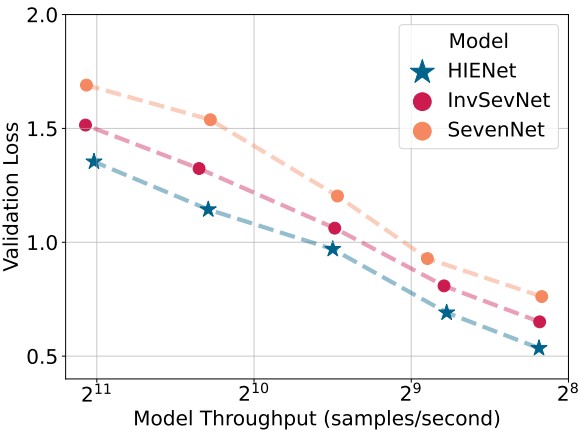

Figure 3: **Hybrid Architecture Ablation**. Model throughput measured on an Nvidia H100 GPU with a batch size of 128. EqvNet uses only equivariant message passing layers and InvNet uses only invariant layers. Validation loss measured on the MPtrj validation set. Hybrid invariant-equivariant models (HIENet) consistently outperform equivariant-only and invariant-only models across all model sizes.

Figure 4: **Hybrid Architecture Generality**. Model throughput measured on an Nvidia H100 GPU with a batch size of 128. InvSevNet represents the SevenNet model (Park et al., 2024b) with an additional invariant message passing layer. InvSevNet outperforms SevenNet across all tested model sizes, but still performs worse than HIENet.

we see that InvNet consistently performs poorly, which aligns with our intuitions that equivariant message passing layers are important to capture high-order atomic interactions and accurately predict force and stress.

**Hybrid architecture generality**. To show that our idea of combining invariant and equivariant layers works well across different model designs, we add our invariant layer to SevenNet to form a hybrid model based on their equivariant designs. As shown in Figure 4, this InvSevNet consistently outperforms the base SevenNet even when controlling for model speed. Additionally, we observe that HIENet still outperforms the InvSevNet model across all tested model sizes. Because our hybrid invariant-equivariant works well with different models, we believe this approach provides a general new direction to design powerful and efficient MLIP models.

## 5.6 Additional Ablation Studies

**O(3) vs. SO(3) equivariant crystal graphs**. To empirically justify why we use O(3) equivariant crystal graph representations instead of the geometrically complete but SO(3) equivariant crystal graphs from Yan et al. (2024), we provide an additional ablation study in Table 6 where we include the additional periodic encoding from Yan et al. (2024). We observe that SO(3) equivariant HIENet performs slightly worse, which aligns with our intuition, since the underlying DFT algorithm is O(3) equivariant.

Table 6: Mean absolute errors on MPtrj validation set for HIENet with O(3) and SO(3) equivariant crystal graphs. Models trained for 20 epochs on the MPtrj dataset. Best performing model in **bold**.

| Equivariance | Energy ↓ (meV/atom) | Force ↓ (meV/Å) | Stress ↓ (kBar) |
|:---:|:---:|:---:|:---:|
| SO(3) | 19.13 | 56.12 | 3.98 |
| O(3) | **16.26** | **49.29** | **3.48** |

**Layer ordering ablation**. Additionally, we investigate different arrangements of message passing layers in Table 7. 'Inv. First' represents our baseline HIENet architecture of applying one invariant layer followed by several equivariant layers, 'Equiv. First' applies several equivariant layers followed by one invariant layer, and 'Mixed' applies alternating invariant and equivariant layers. The invariant-first ordering consistently outperforms other configurations, validating our architectural design choice. We hypothesize that applying invariant layers before equivariant layers builds more informative initial node representations by aggregating topological information (bond lengths, neighbor counts, etc.) from the local chemical environment without incurring the computational overhead of

Table 7: Mean absolute errors on MPtrj validation set for HIENet with different orders of message passing layers. Models trained for 20 epochs on the MPtrj dataset. Best performing model in **bold**.

| MP Layer Ordering | Energy ↓ (meV/atom) | Force ↓ (meV/Å) | Stress ↓ (kBar) |
|---|---|---|---|
| Mixed | 50.35 | 94.62 | 6.41 |
| Equiv. First | 32.26 | 77.22 | 4.94 |
| Inv. First | **16.26** | **49.29** | **3.48** |

tensor products. These initial node representations then give the equivariant layers more informative scalar features to lift into higher-order geometric features, allowing the equivariant layers to better capture angular and high-dimensional physical properties such as force and stress—properties which can only be learned in equivariant layers.

## 6   Conclusion, Limitations, and Societal Impacts

We propose HIENet, a machine learning interatomic potential for materials that demonstrates the importance of (1) integrating invariant and equivariant message-passing layers and (2) satisfying physical constraints for powerful, efficient MLIPs. HIENet outperforms all evaluated baseline methods across a range of benchmarks and applications, while being significantly faster than existing equivariant models. We provide ablation studies to further demonstrate generality and robustness of our hybrid design. Current limitations include (1) focusing primarily on materials discovery, while extensions to other science domains are underexplored, (2) computational constraints preventing training on hundred-million-scale datasets where HIENet's full potential could be realized, and (3) the substantial computational cost of training HIENet makes it infeasible to perform multiple training runs with different random seeds, precluding the reporting of standard deviations for our main results. Future work will explore these directions. The societal impacts of novel materials discovery may apply to this work.

## 7   Broader Impact Statement

The potential benefits and risks associated with AI-powered novel materials discovery may apply to this work.

## 8   Reproducibility Statement

The code for HIENet is available at `https://github.com/divelab/AIRS/tree/main/OpenMat/HIENet` and is included in the supplemental materials to support the reproducibility of the proposed method.

## Acknowledgments

K.Y., M.B., A.K. and S.J. acknowledge partial support from National Science Foundation (NSF) under grant IIS-2243850, ARPA-H under grant 1AY1AX000053, and National Institutes of Health under grant U01AG070112. K.Z., S.K. and X.F.Q. acknowledge partial support from NSF under awards CMMI-2226908 and DMR-2103842 and the donors of ACS Petroleum Research Fund under Grant #65502-ND10. R.A. and D.S. acknowledge support from ARO through Grant No. W911NF-22-2-0117. R.A., X.N.Q. and S.Z. acknowledge NSF through Grant No. 2119103 (DMREF). Z.X. and X.N.Q. acknowledge support from NSF under grants SHF-2215573 and IIS-2212419, and ARPA-H under grant 1AY1AX000053. Some computations were carried out at the Texas A&M High-Performance Research Computing (HPRC) facility. We gratefully acknowledge the support of Lambda Inc. for providing computing resources.

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

# A Molecular Dynamics Simulation

## A.1 Molecular Dynamics Simulations and Structural Optimization of Materials

**Molecular dynamics simulation**. Molecular dynamics (MD) simulation (Alder & Wainwright, 1959) is an important computational method to compute structural, chemical, and thermodynamic properties, which allows for in-depth mechanistic understanding and materials discovery. MD simulation essentially solves Newton's equations of motion for both atomic positions and cell parameters of a material system under a specific thermodynamic ensemble. Specifically, the simulation workflow relies on iterative computation of the total system energy $E$, atomic forces $\mathbf{F}_i$, and stress tensor $\boldsymbol{\sigma}$. For a given starting structure configuration, $E$, $\mathbf{F}_i$, and $\boldsymbol{\sigma}$ can be calculated using classical methods or machine learning interatomic potentials. The acceleration, velocity, and position of atoms can be subsequently determined over a time step through numerical integration methods such as the Velocity-Verlet algorithm under a thermodynamic ensemble. The atomic forces of the new structure will then be updated for the next time step. By iterative numerical integration, the system will evolve under the thermodynamic ensemble and interatomic interactions determined by the force field. Stress also plays a crucial role in MD simulations when controlling pressure, such as in an NPT ensemble (i.e. under the constant number of particle, constant pressure, and constant temperature condition). In order to obtain statistically averaged physical quantities, such calculation needs to be performed iteratively for many time steps, hence computational efficiency becomes critical.

**Structural optimization**. Different from molecular dynamics, structural optimization usually aims to relax the structure and/or cell parameters to their ground state or metastable state. It also involves the calculation of energy, force and stress, which are subsequently used by optimization algorithms or optimizers to update the structure, such as Conjugate Gradient algorithm (CG) (Hestenes et al., 1952) and Broyden–Fletcher–Goldfarb–Shanno algorithm (BFGS) (Fletcher, 2000). This process is repeated until the final convergence criteria is reached.

## A.2 Conventional Computation Methods

Several kinds of simulation techniques are widely used in computational materials science at various scales, such as Density Functional Theory (DFT) (Hohenberg & Kohn, 1964; Kohn & Sham, 1965), MD simulations (Alder & Wainwright, 1959), and Monte Carlo (MC) simulations (Metropolis et al., 1953). DFT is a quantum mechanical method that can be used to simulate material systems at the electronic level. Its key principle is that the ground-state energy of a system can be expressed as a functional of electron density, which reduces $3N_e$-dimensional interacting many-body system down to a fictitious 3-dimensional non-interacting system. However, DFT is computationally expensive and is limited to small systems. MD simulation method has already been elaborated in Appendix A.1 where a force field is required for calculating energy, force, and stress. There are two types of MD simulations depending on the underlying force field: *ab initio* MD (AIMD) simulations where atomic forces are calculated by quantum mechanical method such as DFT, and classical MD simulations where empirical force fields are used to calculate atomic forces. AIMDs are relatively more accurate but computationally expensive, limiting its application to small systems. Classical MD simulations are computationally efficient and can handle large systems, but very often they either lack the accuracy required for highly precise simulations, or cannot be transferred to different simulation conditions. MC simulations are based on statistical mechanics which rely on iterative energy calculations and configuration sampling and updates. Another key challenge is that both classical MD and MC simulations depend on the availability of empirical force fields for the system of interest. Therefore, it is highly desirable to develop machine learning interatomic potentials that can provide accurate and efficient calculations of energy, force, and stress of arbitrary materials system, which will significantly advance materials science, physics and chemistry and allow for studying fundamental mechanism and discovering new materials.

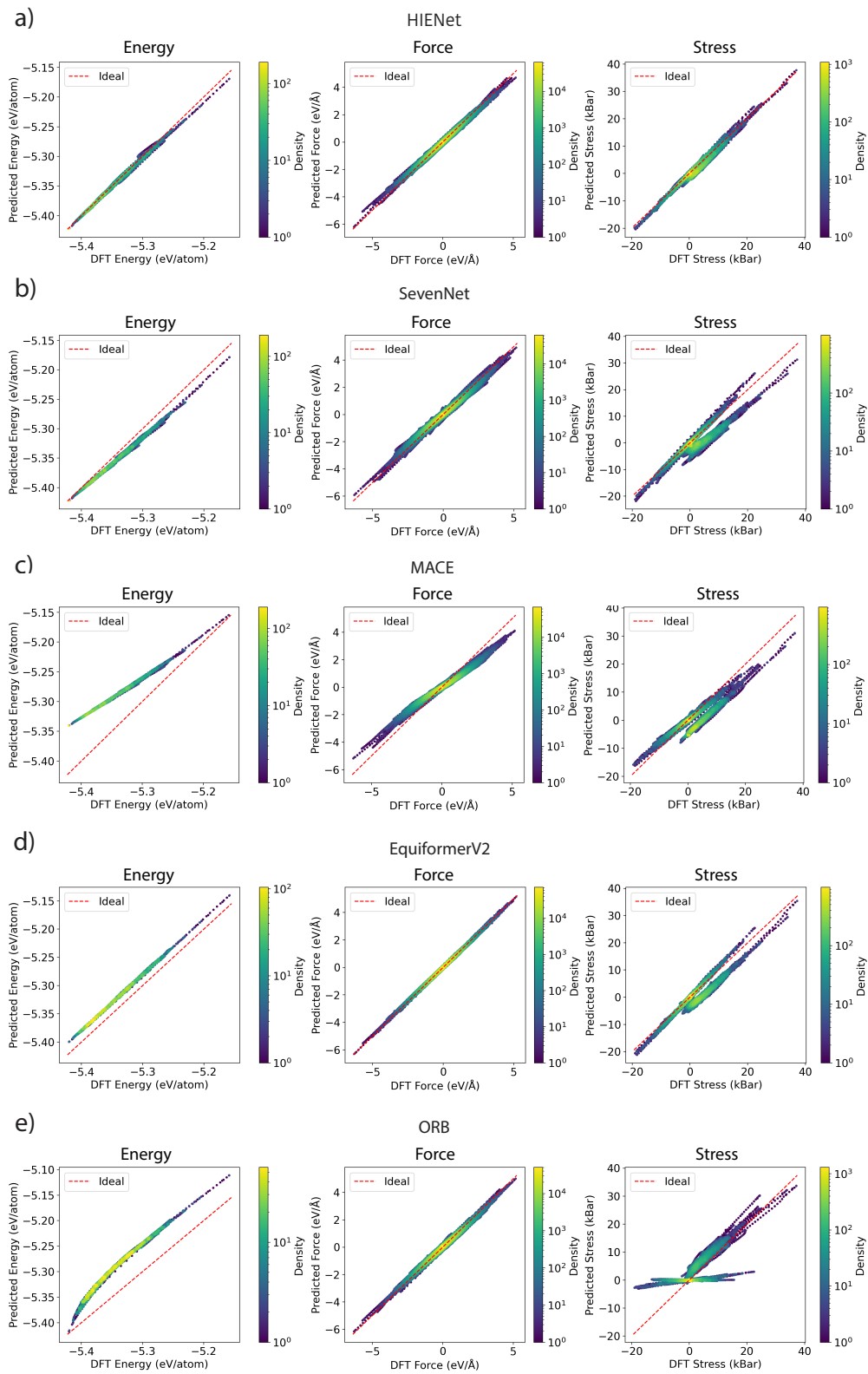

Figure 5: Evaluation of energy, force, and stress predictions for 64-atom Si system calculated by foundation models: a) HIENet, b) SevenNet-l3i5, c) MACE-MP-0, d) CHGNet, and e) eqV2_31M_mp with respect to the DFT results.

Table 8: MLIP prediction accuracy across 4,000 configurations of Si systems. Energy MAE is in meV/atom, Force MAE is in meV/Å, and Stress MAE is in kBar. Best performing model for each metric in **bold** and second best underlined.

|  |  | HIENet | EquiformerV2 | ORB | SevenNet-l3i5 | MACE | CHGNet |
|---|---|---|---|---|---|---|---|
| Energy | MAE | **2** | 19 | 48 | 10 | 55 | 79 |
|  | $R^2$ | **0.995** | 0.794 | -0.344 | 0.932 | -0.762 | -2.617 |
| Force | MAE | 62 | **29** | 65 | 98 | 208 | 267 |
|  | $R^2$ | 0.989 | **0.998** | 0.988 | 0.974 | 0.885 | 0.805 |
| Stress | MAE | **0.995** | 2.065 | 2.693 | 2.171 | 2.918 | 4.224 |
|  | $R^2$ | **0.938** | 0.706 | 0.507 | 0.652 | 0.466 | -0.102 |

## B   Evaluations on *Ab Initio* Molecular Dynamics

As mentioned in Appendix A.1, *Ab initio* molecular dynamics (AIMD) simulation is an incredibly important application of machine learning interatomic potentials (MLIPs). Here we evaluate HIENet and baseline MLIPs on AIMD simulations.

To evaluate MLIPs performance, we generate a testing dataset consisting of silicon (Si) systems containing 64 atoms in a $2 \times 2 \times 2$ supercell. A Γ-centered Monkhorst–Pack k-point sampling grid of $2 \times 2 \times 2$ (Monkhorst & Pack, 1976) was used. AIMD simulations were performed in the NVT ensemble with a Nosé-Hoover thermostat at four temperatures of 300, 500, 700, and 900 K with time step of 1 fs for 1,000 steps at each temperature. In total, 4,000 configurations were generated for model evaluation. Since configurations are sampled from a variety of temperatures, this task represents an out-of-distribution generalization problem compared to the MPtrj training dataset. We select Si systems because it is a representative material of great interest and importance to the semiconductor industry.

AIMD simulations were conducted using DFT as implemented in VASP with the PBE exchange-correlation energy functional. A plane-wave basis set with a cutoff energy of 520 eV was used to ensure numerical accuracy in the simulations. To ensure consistency between training and evaluation, all input settings were generated using the MPRelaxSet class.

For each system configuration, we compute MLIP energy, forces, and stress and compare with DFT reference data. As shown in Table 8, HIENet achieves vastly better accuracy on energy and stress performance compared to baseline models, though EquiformerV2 has better accuracy on force predictions. Parity plots for each model are shown in Fig. 5, where we observe that HIENet consistently performs well across all system configurations.

## C   Evaluations on Alloys

We also evaluate MLIP performance on phase diagram calculations using the Alloy Theoretic Automated Toolkit (ATAT) (Van De Walle et al., 2002) framework following the approach outlined in Zhu et al. (2025). Phase diagrams are graphical representations of the state of materials under arbitrary conditions and accurately predicting them is a necessary condition for the further development of complex materials (Arróyave, 2022).

Starting with the simple Au-Pt binary systems, we first generate Special Quasirandom Structures (SQS) (Zunger et al., 1990) of FCC Au-Pt with different compositions using ATAT, with 32 atoms in a $2 \times 2 \times 2$ supercell—the SQS structures are designed to mimic disordered alloys within a certain precision. Then, the relaxation and free energy calculations are carried out using ab initio calculations and MLIPs. For all *ab initio* calculations, VASP (Kresse & Hafner, 1993; 1994; Kresse & Furthmüller, 1996a;b) is used with the PBE exchange-correlation functional and PAW pseudopotentials at the level of GGA (Blöchl, 1994; Perdew et al., 1996). The k-point density is set to 8,000 k-points per reciprocal atom for all calculations.

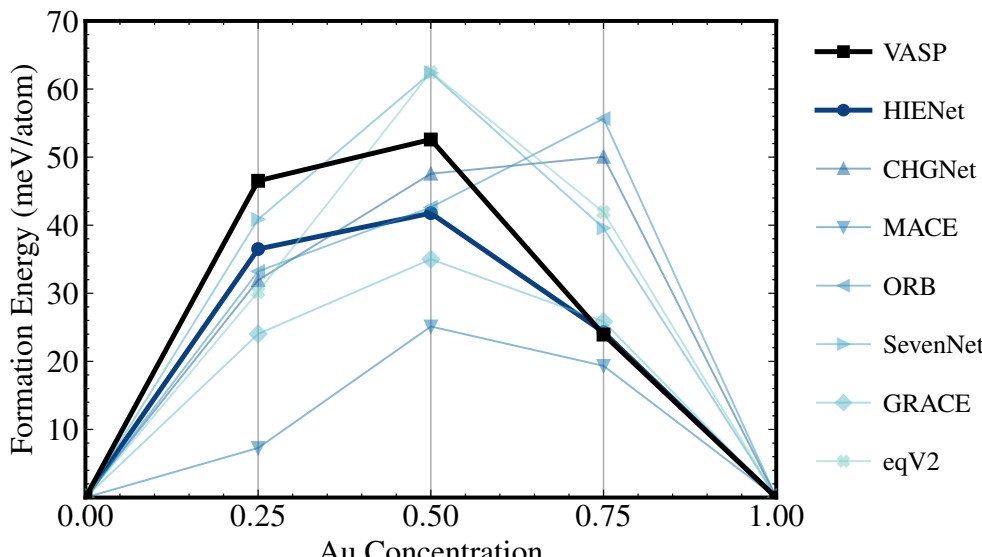

Figure 6: Formation energies per atom of the Au-Pt binary FCC system calculated with models trained on the MPtrj dataset. eqV2 refers to EquiformerV2 and SevenNet is the SevenNet-l3i5 model.

Table 9: Ordering of Au-Pt formation energies calculated with different potentials (1 for the lowest formation energy and 3 for the highest). Ideally, MLIP predictions should match the VASP ordering of formation energies.

| Model | Ordering of Formation Energies | | |
|---|---|---|---|
| | $\Delta G\left(x_{\mathrm{Au}} = 0.25\right)$ | $\Delta G\left(x_{\mathrm{Au}} = 0.5\right)$ | $\Delta G\left(x_{\mathrm{Au}} = 0.75\right)$ |
| CHGNet | 1 | 2 | 3 |
| MACE | 1 | 3 | 2 |
| ORB | 1 | 2 | 3 |
| SevenNet-l3i5 | 2 | 3 | 1 |
| GRACE | 1 | 3 | 2 |
| EquiformerV2 | 1 | 3 | 2 |
| HIENet | 2 | 3 | 1 |
| VASP | 2 | 3 | 1 |

In Figure 6, we plot the formation energies of the Au-Pt FCC binary systems calculated by HIENet and baseline MLIPs. We see that HIENet shows strong agreement with first-principles DFT results as our model predictions closely match the true formation energy across all Au concentrations.

In addition, although all the models successfully give a positive formation energy for the SQS's, predicting the miscibility gap in the phase diagram, most of the models including CHGNet, MACE, ORB, GRACE and EquiformerV2 fail to reproduce the correct ordering of the formation energies: $\Delta G\left(x_{\mathrm{Au}} = 0.5\right) > \Delta G\left(x_{\mathrm{Au}} = 0.25\right) > \Delta G\left(x_{\mathrm{Au}} = 0.75\right)$, as shown in Table 9. Such ordering of formation energies is highly important in thermodynamics and materials science, as it governs the stability of the phases.

Finally, we demonstrate how HIENet can be used for multi-element systems. In Figure 7, we present a ternary phase diagram for the Cr-Mo-V system at 1,000 K calculated with ATAT and HIENet. The ternary phase diagram calculation correctly identifies the BCC phase miscibility gap in the Cr-Mo region.

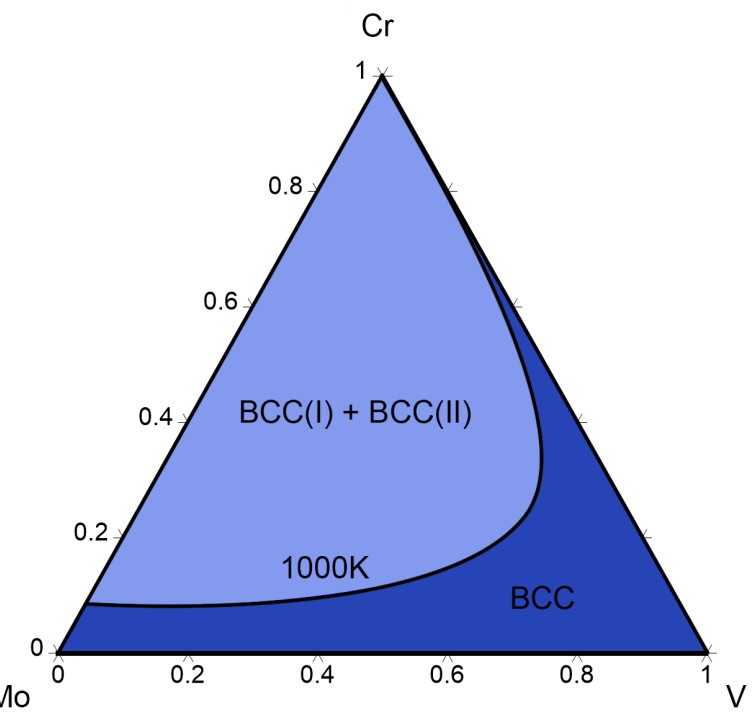

Figure 7: Cr-Mo-V ternary phase diagram at 1,000 K calculated with ATAT and HIENet. Only the BCC phase is included in the calculation. The phase diagram is plotted with the Pandat (Chen et al., 2002) software package.

## D Evaluations on Phonon and Bulk Modulus

### D.1 Phonon Frequency Evaluation

As the calculations of the Material Project phonon dataset were performed using the PBEsol exchange-correlation energy functional, it would be inconsistent to compare them with the models trained on the data using the Perdew-Burke-Ernzerhof (PBE) (Perdew et al., 1996) exchange-correlation energy functional. PhononDB, a database of phonon calculations including band structure, DOS, and thermal properties for over 10,000 materials evaluated using the PBE functional, provides a more effective reference for comparison, hence was used as the reference for the evaluation as detailed below. Phonon frequencies and corresponding band structures were computed using the Phonopy package via the finite displacement method (Togo et al., 2023; Togo, 2023) where MLIPs were employed to compute the dynamical matrices and corresponding phonon band structures of each crystal structure. To ensure direct comparison between PhononDB and calculated data, the Phonopy objects were initialized with the same unit cell and supercell matrices as used in PhononDB calculations. Additionally, the primitive cell matrix was included if defined. Displaced supercells were generated using a default displacement of 0.01 Å and the corresponding forces were evaluated with our model. High-symmetry k-path in the Brillouin zone was computed using SeeK-Path (Hinuma et al., 2017; Atsushi Togo & Tanaka, 2024). Using this workflow, the high-symmetry k-paths and the sampling grids were identical between the reference phonon band structure from PhononDB and the predicted band structure from our model.

In addition to the frequency evaluation in Table 3, we provide several phonon band structure diagrams calculated using HIENet in Figure 8 for Si, CdTe, $Cs_2KInF_6$, and $GaAgS_2$ systems. We observe that the HIENet-predicted phonon band structures of Si exhibits reasonable accuracy, and the phonon band structures for CdTe, $Cs_2KInF_6$, and $GaAgS_2$ are in very good agreement with the PhononDB DFT results across the entire frequency range and high-symmetry k-paths. Furthermore, the phonon band structure of $Cs_2KInF_6$ contains negative phonon frequencies, indicating the dynamical instability of the crystal structure despite its

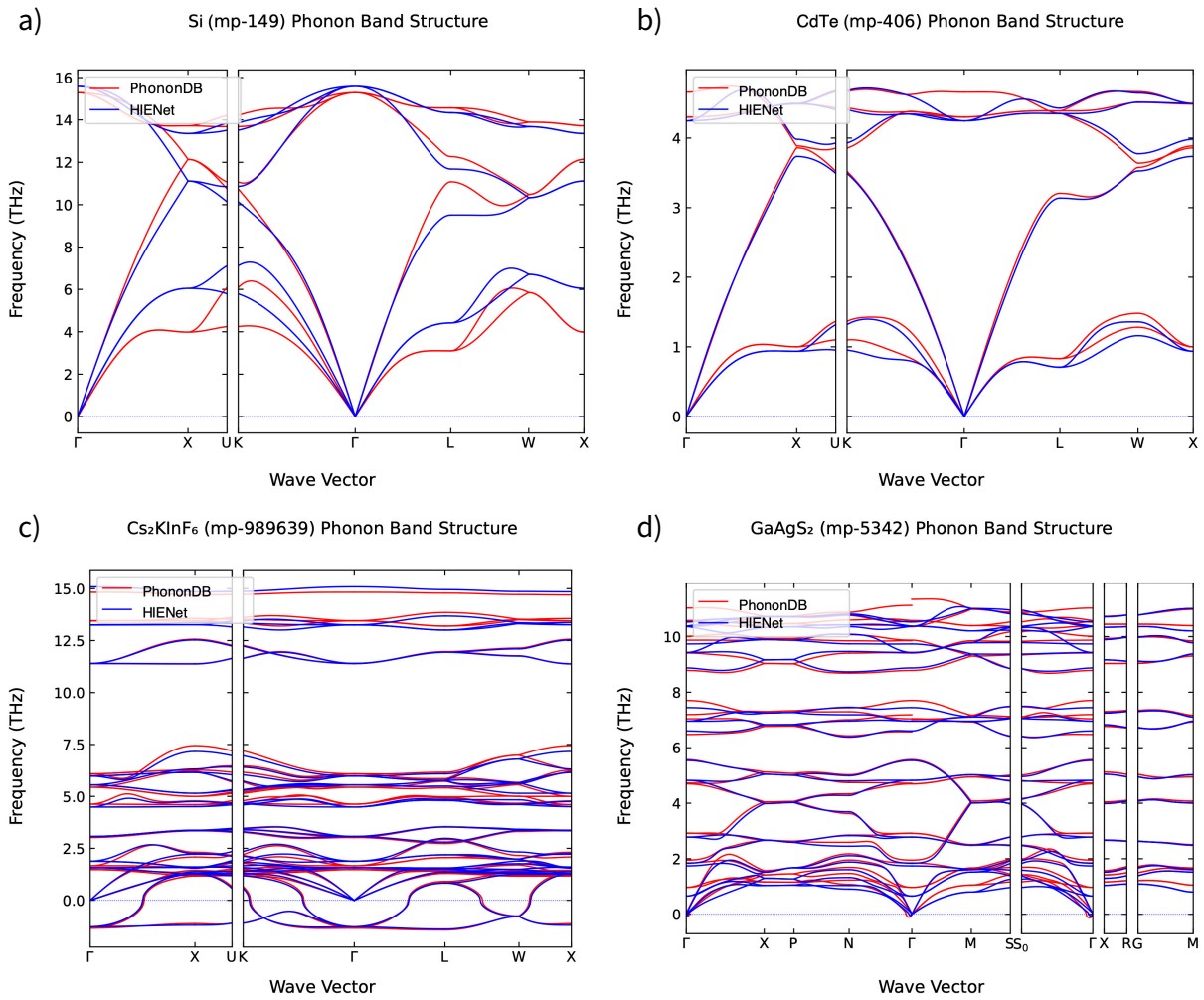

Figure 8: Phonon band structures for a) Si, b) CdTe, c) $Cs_2KInF_6$, and d) $GaAgS_2$ calculated using HIENet compared with reference data in the PhononDB database.

local stability. Impressively, the HIENet predictions agree with the DFT data extremely well even in this negative frequency regime across all high-symmetry pathways. As such, HIENet can be a powerful MLIP for predicting a materials thermal conductivity and structural stability.

## D.2   Bulk Modulus Evaluation

To compute bulk modulus, we need to calculate the elastic tensor for each crystal. The latter is calculated by first relaxing the input structure to the default force tolerance of 1.1 eV/Å using each MLIP. The relaxed structure is then deformed with strains of ($\pm 0.005$, $\pm 0.01$) applied to normal modes and strains of ($\pm 0.06$, $\pm 0.03$) applied to shear modes for a total of 4 strain magnitudes for each of the 6 strain modes. The resulting stress-strain values are fit linearly to compute the elastic tensor. The reference elastic constants in the Materials Project were calculated using DFT with the PBE functional in the generalized gradient approximation (GGA) (Langreth & Mehl, 1983) as implemented the Vienna Ab-initio Simulation Package (VASP) (Kresse & Furthmüller, 1996b). For metallic entries, a plane wave cutoff energy of 700 eV with k-point density of 7,000 per reciprocal atom was used. For non-metallic entries such as insulators or semiconductors, a plane wave cutoff energy of 700 eV was once again used with a k-point density of 10,000 per reciprocal

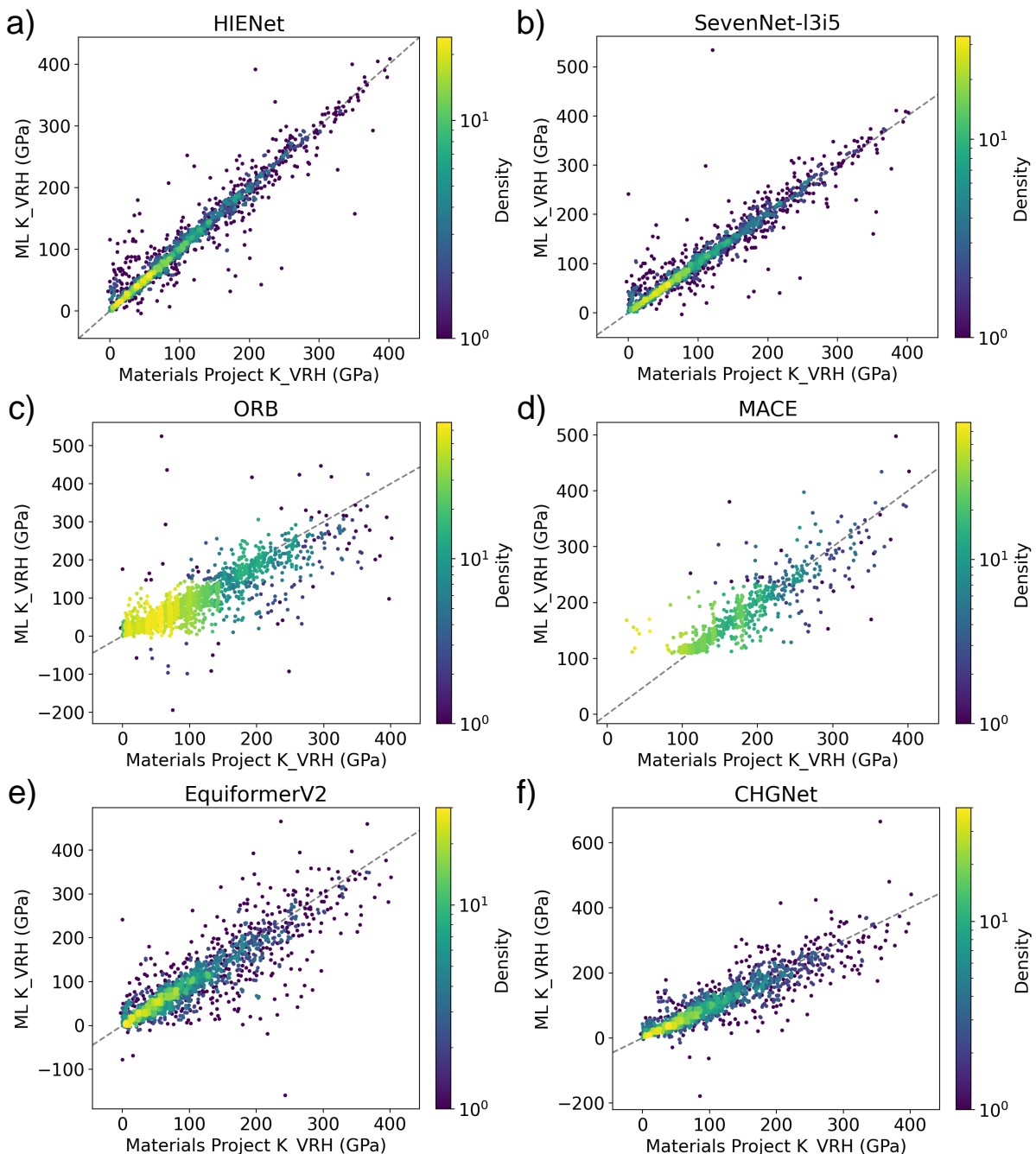

Figure 9: Comparison of bulk modulus $K_{VRH}$ calculated by a) HIENet, b) SevenNet-l3i5, c) ORB, d) MACE, e) EquiformerV2, and f) CHGNet with the reference data in the Materials Project database.

atom (De Jong et al., 2015). In addition to the main results reported in Table 4, we provide parity plots for each model in Fig. 9.

# E    Recent MLIP Developments

Several new MLIPs trained on the MPtrj dataset have been proposed that achieve strong performance on Matbench Discovery, which are summarized in Table 10. eSEN (Fu et al., 2025) is an equivariant model that shares our emphasis on physical constraints, using conservative gradient-based force predictions and polynomial envelope functions. MatRIS (Zhou et al.) and DPA-3.1 (Zhang et al., 2025) are purely invariant models that achieve strong performance through expressive three-body interaction modeling via line graph construction, demonstrating that the invariant-only paradigm remains a promising direction when equipped with sufficiently expressive geometric encodings. Nequix (Koker et al., 2025) are lightweight equivariant models that achieve competitive performance with remarkably few parameters. GRACE-2L (Lysogorskiy et al., 2026) is an equivariant model based on atomic cluster expansion with conservative gradient-based predictions. Eqnorm (Chen, 2025) employs a dual-branch design combining equivariant tensor features with invariant scalar embeddings, independently corroborating our central finding that hybrid invariant-equivariant architectures offer an effective balance between expressiveness and efficiency.

We note that the primary contribution of this work is not to claim state-of-the-art performance, but to demonstrate that the hybrid invariant-equivariant design is a powerful and general architectural paradigm, and that enforcing physical constraints is essential for robust downstream performance. We believe the hybrid design could be readily applied to further boost these recent architectures. In particular, prepending an invariant layer to equivariant models such as eSEN and Nequix could provide further performance gains at minimal computational cost, as demonstrated by our InvSevNet experiments in Sec. 5.5.

Table 10: Model performance on the Unique Prototype split of the Matbench Discovery benchmark for recent MPtrj-trained models. DAF is the Discovery Acceleration Factor. MAE and RMSE are in meV/atom. Missing results marked by -.

| Model | HIENet | eSEN-30M-MP | MatRIS-10M-MP | DPA-3.1-MPtrj | Eqnorm MPtrj | Nequix MP | GRACE-2L-MPtrj |
|---|---|---|---|---|---|---|---|
| DAF ↑ | 4.932 | _5.260_ | **5.422** | 5.024 | 4.844 | 4.455 | 4.163 |
| MAE ↓ | 41 | _33_ | **31** | 37 | 40 | 44 | 52 |
| RMSE ↓ | 84 | _78_ | **77** | 80 | 83 | 86 | 94 |
| $R^2$ ↑ | 0.793 | _0.822_ | **0.824** | 0.812 | 0.799 | 0.782 | 0.741 |
| F1 ↑ | 0.777 | _0.831_ | **0.847** | 0.803 | 0.786 | 0.751 | 0.691 |
| Accuracy ↑ | 0.929 | _0.946_ | **0.951** | 0.936 | 0.929 | 0.914 | 0.895 |
| Precision ↑ | 0.754 | _0.804_ | **0.829** | 0.768 | 0.741 | 0.681 | 0.636 |
| RMSD ↓ | 0.080 | _0.075_ | **0.072** | 0.080 | 0.084 | 0.085 | 0.090 |
| Params | 7.51M | 30.1M | 10.4M | 4.81M | 1.31M | 708k | 15.3M |

# F    Hyperparameter Sensitivity Analysis

Table 11: Hyperparameter sensitivity analysis on MPtrj validation set. Models trained for 10 epochs. Best performing model in **bold** and second best _underlined_.

| Cutoff (Å) | $L_{max}$ | Speed ↑ (samples/sec) | Energy ↓ (meV/atom) | Forces ↓ (meV/Å) | Stress ↓ (kBar) |
|---|---|---|---|---|---|
| 4 | 3 | 17.10 | 16.27 | 45.51 | 3.31 |
| 5 | 2 | **18.75** | 14.16 | 44.12 | 3.21 |
| 5 | 3 | 15.58 | 13.92 | 42.83 | _3.19_ |
| 5 | 4 | 10.64 | _12.55_ | _40.19_ | **3.06** |
| 6 | 3 | 12.52 | **12.95** | **40.86** | 3.12 |

Here we evaluate the sensitivity of HIENet to key architectural hyperparameters. As shown in Table 11, increasing the cutoff radius and maximum spherical harmonics order $L_{max}$ consistently improves model

accuracy but reduces inference speed. The chosen combination for the final model of 5Å cutoff radius and $L_{\max} = 3$ provides a good balance between accuracy and computational efficiency. All tested configurations demonstrate stable training convergence.

## G   Model Settings and Experimental Details

### G.1   HIENet Settings

HIENet consists of 1 invariant and 3 equivariant message passing layers. For the invariant message passing layers, we use a hidden dimension of 512 for node features and a single attention head. The equivariant layers use a representation that consists of 512 scalar channels with $l = 0$, 128 vector channels with $l = 1$, 64 higher-order tensor channels with $l = 2$, and 32 higher-order tensor channels with $l = 3$. We use 8 radial Bessel basis functions for distance encoding and a polynomial envelope (Gasteiger et al., 2020) with $p = 6$. We use SiLU and sigmoid activation functions (Elfwing et al., 2018) throughout the network to ensure smooth and continuously differentiable gradients. To prevent overfitting, we regularize the model by applying dropout to the MLPs that operate on scalar features in both invariant and equivariant message passing layers. Specifically, we employ a dropout rate of $p_{\text{attn}} = 0.1$ for MLPs involved in attention calculations, while using a lower rate of $p = 0.06$ for all other MLPs in the network. Additionally, we scale the input energies by the root mean square (RMS) of forces from the training dataset and shift by element-wise reference energies from the same dataset.

Following Batatia et al. (2023), we split the Materials Project Trajectory (MPtrj) Dataset (Deng et al., 2023a) into training (95%) and validation (5%) sets. We train the model for 250 epochs on a platform with 2 AMD EPYC 7J13 64-Core Processors (240 cores total), 1.7 TiB DDR4 memory, and 8 NVIDIA A100-SXM4-80GB GPU accelerators. We use a total batch size of 384 (48 per GPU), which results in the model taking 118 minutes per training epoch and 6 minutes per validation epoch.

We provide the code used for training in the supplementary materials.

### G.2   Optimization

We optimize HIENet using the AdamW optimizer (Loshchilov & Hutter, 2019) with weight decay of 0.001. The learning rate follows a cosine annealing schedule (Loshchilov & Hutter, 2022) with an initial warm-up phase to stabilize early training.

The loss function combines energy, force, and stress predictions with different weighting factors as:

$$\mathcal{L} = \lambda_E \mathcal{L}_E + \lambda_F \mathcal{L}_F + \lambda_\sigma \mathcal{L}_\sigma \tag{13}$$

where $\mathcal{L}_E$, $\mathcal{L}_F$, and $\mathcal{L}_\sigma$ represent the Huber losses for energy, force, and stress predictions, respectively, with $\delta = 0.01$. We set the weighting coefficients $\lambda_E = 1.0$, $\lambda_F = 1.0$, and $\lambda_\sigma = 0.01$.

To improve model generalization and training stability, we additionally maintain an exponential moving average (EMA) of model parameters with a decay rate of 0.999.

The hyperparameters for both the model architecture and optimization are summarized in Table 12.

### G.3   Envelope Function

As mentioned in Sec. 4.1, we use the polynomial envelope function (Gasteiger et al., 2021):

$$f_{\text{poly}}(r) = 1 - \frac{(p+1)(p+2)}{2}d^p + p(p+2)d^{p+1} - \frac{p(p+1)}{2}d^{p+2} \tag{14}$$

where $p \in \mathbb{Z}, 0 < p$. In practice, we select $p = 6$. It is critical to have such an envelope function in order to ensure that the MLIP energy predictions are continuously differentiable with respect to atom positions. The polynomial envelope was selected because the first and second derivatives of $\boldsymbol{h}_{ji}$ will then go to 0 at the cutoff radius $R_{\max}$.

Table 12: Hyperparameters used for model training.

| Hyperparameter | Value |
|---|---|
| Optimizer | AdamW |
| Learning rate scheduler | Cosine Annealing |
| Maximum learning rate | 0.01 |
| Minimum learning rate | 0.000005 |
| Warmup epochs | 0.1 |
| Warmup factor | 0.2 |
| Number of epochs | 250 |
| Batch size | 48 |
| Weight decay | 0.001 |
| Dropout rate, $p$ | 0.06 |
| Attention dropout rate, $p_{\text{attn}}$ | 0.1 |
| Energy loss weight, $\lambda_E$ | 1.0 |
| Force loss weight, $\lambda_F$ | 1.0 |
| Stress loss weight, $\lambda_\sigma$ | 0.01 |
| Model EMA Decay | 0.999 |

## H  Detailed Equivariance Proof

Here we provide a more rigorous and detailed proof of the Proposition 4.3 that pertains to the $O(3)$ equivariance of HIENet's predictions.

*Proof of Proposition 4.3.* First, the radius-based graph construction $\vartheta_{\text{graph}}$ described in Sec. 4.1 is $O(3)$ equivariant:

$$\vartheta_{\text{graph}}(\mathbf{Z}, \mathbf{RP} + \mathbf{b}, \mathbf{RL}) = \mathbf{R}\vartheta_{\text{graph}}(\mathbf{Z}, \mathbf{P}, \mathbf{L})$$

This is because the radius-based graph construction only depends on the relative positions between atoms and the resulting displacement vectors $\boldsymbol{r}_{ji}$ will rotate accordingly.

Second, the proposed HIENet message passing layers $\vartheta_{\text{HIENet}}$ are $E(3)$ invariant for the final energy prediction. The invariant message passing layers are $E(3)$ invariant by construction because they only operate on the magnitude $||\boldsymbol{r}_{ji}||$ and for the equivariant message passing layers, we only extract the final $l = 0$ features, which are invariant by the definition of the Clebsch-Gordan tensor product. Because of this:

$$\hat{E}(\mathbf{Z}, \mathbf{RP} + \mathbf{b}, \mathbf{RL}) = \vartheta_{\text{HIENet}}(\vartheta_{\text{graph}}(\mathbf{Z}, \mathbf{RP} + \mathbf{b}, \mathbf{RL})) = \vartheta_{\text{HIENet}}(\mathbf{R}\vartheta_{\text{graph}}(\mathbf{Z}, \mathbf{P}, \mathbf{L})) \tag{15}$$

$$= \vartheta_{\text{HIENet}}(\vartheta_{\text{graph}}(\mathbf{Z}, \mathbf{P}, \mathbf{L})) = \hat{E}(\mathbf{Z}, \mathbf{P}, \mathbf{L}), \tag{16}$$

Therefore, HIENet energy predictions are $E(3)$ invariant. Based on the physics informed property predictions described in Sec. 4.3, we then have:

$$\hat{\boldsymbol{F}}_i(\mathbf{Z}, \mathbf{RP} + \mathbf{b}, \mathbf{RL}) = -\nabla_{\mathbf{R}\boldsymbol{p}_i}\hat{E}(\mathbf{Z}, \mathbf{P}, \mathbf{L}) = -\mathbf{R}\nabla_{\boldsymbol{p}_i}\hat{E}(\mathbf{Z}, \mathbf{P}, \mathbf{L}) = \mathbf{R}\hat{\boldsymbol{F}}_i(\mathbf{Z}, \mathbf{P}, \mathbf{L}), \tag{17}$$

$$\hat{\boldsymbol{\sigma}}(\mathbf{Z}, \mathbf{RP} + \mathbf{b}, \mathbf{RL}) = \frac{1}{V}\nabla_{\mathbf{R}\epsilon_{ij}\mathbf{R}^T}\hat{E} = \frac{1}{V}\mathbf{R}\nabla_{\epsilon_{ij}}\hat{E}\mathbf{R}^T = \mathbf{R}\hat{\boldsymbol{\sigma}}(\mathbf{Z}, \mathbf{P}, \mathbf{L})\mathbf{R}^T, \tag{18}$$

therefore energy, force, and stress each transform appropriately under rototranslation and HIENet predictions are $O(3)$ equivariant. $\square$

## I  LLM Usage

We have used LLMs to polish our paper writing. Specifically, we have used LLMs to refine wording and grammar throughout the paper. The research contributions, experimental design, analysis, and conclusions are our own work.

## J    Licenses for Existing Assets

We have used datasets including the Materials Project Trajectory (MPtrj) dataset (Deng et al., 2023a) with MIT License and Materials Project Database (Jain et al., 2013) with the Creative Commons Attribution 4.0 International License. For evaluations, we have used the Matbench Discovery benchmark (Riebesell et al., 2023) with MIT License, Phonopy and Togo PhononDB Database (Togo et al., 2023; Togo, 2023) with the BSD 3-Clause License, Alloy Theoretic Automated Toolkit (ATAT) (Van De Walle et al., 2002) with the Creative Commons Attribution-NoDerivatives 4.0 International License, and MatCalc's Elasticity module (Liu et al., 2024) with the BSD 3-Clause License. For model comparisons, we included EquiformerV2 (Liao et al., 2024) with the MIT License, ORB (Neumann et al., 2024) with the Apache License Version 2.0, SevenNet (Park et al., 2024b) with the GNU General Public License Version 3.0, GRACE (Bochkarev et al., 2024) with the Academic Software Licence, MACE (Batatia et al., 2023) with the MIT License, and CHGNet (Deng et al., 2023b) with the BSD 3-Clause License.

