# OpenReview forum: "Learning Materials Interatomic Potentials via Hybrid Invariant-Equivariant Architectures"
_TMLR — Accepted by TMLR_

### Review · Reviewer_preZ · 2026-02-13

**Summary Of Contributions:**

The paper introduces a novel neural network architecture designed for machine learning interatomic potentials. It emphasizes the importance of adhering to physical constraints, such as equivariance, in predicting forces, energies, and stresses. The proposed architecture combines a gated graph transformer for invariant atomic features with equivariant features updated via tensor products involving spherical harmonic representations of edge vectors. The model is evaluated on the Matbench Discovery benchmark, the Materials Project Trajectory dataset, phonon frequency predictions, and elastic property predictions and performs well across these tasks.

The paper is clear and readable, with most details explained sufficiently, except for some missing information regarding the ablation study. A few claims should be revised, in my opinion, as noted below. While the paper references relevant literature, it does not cite some of the most recent works from 2025, and key new results on the Matbench Discovery benchmark are not included for comparison.

**Audience:**

Yes

**Audience Explanation:**

The paper presents a solid study, proposing an architecture that is both reasonable and effective, though it may not fully match the performance of some recently introduced models. However, its simplicity is a notable strength. The emphasis on adhering to physical constraints is significant and well-justified, even if the methods employed are not entirely novel. These aspects are clearly explained in the paper. The experiments are thorough, with particularly interesting results on phonon frequencies and elastic properties.

**Broader Impact Concerns:**

I have no concerns

**Claims And Evidence:**

No

**Claims Explanation:**

"Moreover, in contrast to existing models [...] methods.
The sentence is unclear about which physical constraints are fulfilled or not fulfilled in models such as Equiformer V2 and Orb. Also, several other models adhere to physical constraints by enforcing pyhisical symmetries and conservation laws.

"[...] HIENet achieves improved performance [...] prior models."
Could you clarify which baselines you're referring to?

"[...] satisfies all desirable [...]" (two times)
The claim is too broad.

"[...] such models under-perform [...]"
This claim is too general in my view - there surely are MLIPs that perform well on such tasks.

"Unlike previous work [...] our constructed crystal graphs are O(3) equivariant [...]"
This method of constructing crystal graphs appears to be quite standard, though the sentence suggests it is being presented as a novel or unique feature.

"One important point missing from the existing literature [...]"
This is also mentioned e.g. in Fu et al. "Learning Smooth and Expressive Interatomic Potentials for Physical Property Prediction".

"Moreover, even some equivariant models [...] violate force conservation [...]"
I think this should rather be phrased in terms of models that employ direct force prediction. It is trivial to equip an EquiformerV2 potential with energy-conserving force predictions by differentiation.

Please include the most recent, best performing models as comparisons in all experiments.

**Requested Changes:**

Required changes:
- Kindly address all the outlined claims as specified above.

Non-critical changes:

- Shouldn't Z be in Z^n (integer) instead of R^n?

- I think the notation around Eq. 7 could be clarified. Specifically, please define TP_l explicitly (I assume it refers to a projection of the Clebsch-Gordan tensor product?) Could this not be written more clearly with a standard C-G tensor product? Additionally, clarify the meaning of Wf_i in this context. Is it intended to represent a per-order linear mixing?

- Regarding proposition 3.5 (force equilibrium), I believe this also follows directly from translational invariance of the energy.

- Additional details about the ablations in Section 5.5 would be helpful. How exactly are the ablated models and the modified SevenNet models constructed? Figure 3 and Figure 4 are somewhat unclear to me. I assume you vary model sizes (how exactly?) to achieve different levels of throughput for the same architecture, and then measure validation loss (which specific metric is used?). Would it be possible to use one or more of the same metrics as presented in Table 2?

---

> ### Author Response · Authors · 2026-03-17
>
> We thank the reviewer for their thoughtful feedback and careful reading of the paper. We address each raised point below.
>
> > Please include the most recent, best performing models as comparisons in all experiments.
>
> We thank the reviewer for this suggestion. We acknowledge that several recent MLIPs have been proposed that achieve strong performance on Matbench Discovery and related benchmarks. We have provided citations and discussion of these models in the appendix of the revised manuscript. We note that the primary contribution of this work is not to train the most powerful possible foundation model, as such exploration would require an infeasible computational budget and involve training on larger datasets such as OMAT. Rather, our object is to demonstrate that our proposed hybrid invariant-equivariant design is a powerful and general architectural paradigm for MLIPs and to study the importance of satisfying physical constraints. These contributions are orthogonal to the performance of any specific model, and we believe the hybrid design could be applied to further boost the performance of more recent architectures, as demonstrated by our InvSevNet experiments in Sec. 5.5.
>
> For completeness, we provide the comparison with more recent models on the Matbench Discovery benchmark (unique structure prototypes split, compliant MPtrj-trained models only) below:
>
> | Model | F1 ↑ | DAF ↑ | MAE ↓ | RMSE ↓ | R² ↑ | Acc ↑ | Prec ↑ | RMSD ↓ | Params |
> |---|---|---|---|---|---|---|---|---|---|
> | HIENet | 0.777 | 4.932 | 41 | 84 | 0.793 | 0.929 | 0.754 | 0.080 | 7.51M |
> | Eqnorm MPtrj | 0.786 | 4.844 | 40 | 83 | 0.799 | 0.929 | 0.741 | 0.084 | 1.31M |
> | DPA-3.1-MPtrj | 0.803 | 5.024 | 37 | 80 | 0.812 | 0.936 | 0.768 | 0.080 | 4.81M |
> | eSEN-30M-MP | 0.831 | **5.260** | **33** | **78** | **0.822** | **0.946** | **0.804** | **0.075** | 30.1M |
> | MatRIS-10M-MP | **0.847** | 5.422 | 31 | 77 | 0.824 | 0.951 | 0.829 | 0.072 | 10.4M |
>
> > "Moreover, in contrast to existing models [...] methods." The sentence is unclear about which physical constraints are fulfilled or not fulfilled in models such as EquiformerV2 and Orb.
>
> We have revised the sentence to explicitly state what physical constraints each model satisfies and violates. Specifically, EquiformerV2 enforces O(3) equivariance but violates force conservation through direct force prediction, while ORB learns rotational invariances from data rather than enforcing them architecturally.
>
> > "[...] HIENet achieves improved performance [...] prior models." Could you clarify which baselines you're referring to?
>
> The baselines we refer to are EquiformerV2, SevenNet, MACE, ORB, and CHGNet. For more recently proposed models, we direct the reviewer to our response to the "Please include the most recent models" point above, as well as the dedicated discussion in Appendix E of the revised manuscript. We have revised the sentence to explicitly list the baselines for clarity.
>
> > "[...] satisfies all desirable [...]" (two times) The claim is too broad.
>
> We have revised both instances to reference the specific constraints formally defined in Sec. 3.1, making the claim more precise.
>
> > "[...] such models under-perform [...]" This claim is too general in my view - there surely are MLIPs that perform well on such tasks.
>
> We agree the original phrasing was a bit too broad and have revised it accordingly. The claim specifically refers to models that violate key physical constraints, such as EquiformerV2 and ORB, which despite achieving competitive results on standard energy/force benchmarks, perform significantly worse on downstream tasks requiring physically meaningful predictions. This is directly supported by our phonon frequency and bulk modulus evaluations in Sec. 5.3, where EquiformerV2 and ORB have noticeably higher errors than models that satisfy physical constraints.

---

> ### Author Response · Authors · 2026-03-17
>
> We continue addressing the reviewer's points below.
>
> > "Unlike previous work [...] our constructed crystal graphs are O(3) equivariant [...]" This method of constructing crystal graphs appears to be quite standard, though the sentence suggests it is being presented as a novel or unique feature.
>
> We agree with the reviewer and have revised the paragraph accordingly. While radius-based graph construction is indeed standard, the key point we aim to highlight is that some prior crystal graph works [1] additionally incorporate SO(3) equivariant periodic encodings to achieve geometric completeness, which however breaks O(3) equivariance. Since O(3) equivariance of the input crystal graphs is a necessary condition for achieving O(3) equivariant MLIP predictions, and since the underlying DFT calculations are inherently O(3) equivariant, this introduces a symmetry mismatch. We empirically verify this in Sec. 5.6, where we show that using SO(3) equivariant graph encodings leads to worse MLIP performance. The revised paragraph makes clear that the novelty is not in the graph construction itself, but in this analysis on the importance of preserving O(3) equivariance throughout.
>
> > "One important point missing from the existing literature [...]" This is also mentioned e.g. in Fu et al. "Learning Smooth and Expressive Interatomic Potentials for Physical Property Prediction".
>
> We thank the reviewer for this reference. We have added a citation to Fu et al. and revised the sentence in the final manuscript to acknowledge it.
>
> > "Moreover, even some equivariant models [...] violate force conservation [...]" I think this should rather be phrased in terms of models that employ direct force prediction.
>
> We have revised the sentence to be more precise. We note that while gradient-based force prediction is necessary for force conservation, it is not sufficient—the model must be carefully designed throughout to ensure the predicted energy is continuously differentiable with respect to atom positions. For example, smooth cutoff functions must be used as we discuss in Sec. 4.1. While some models such as EquiformerV2 can indeed be equipped with gradient-based force predictions to achieve force conservation, evaluating arbitrary equivariant models requires careful attention to the entire model design, not just the prediction head.
>
>
> > Shouldn't Z be in Z^n (integer) instead of R^n?
>
> We thank the reviewer for catching this. The atomic numbers $\mathbf{Z}$ are indeed integers and we have corrected the notation to $\mathbf{Z} \in \mathbb{Z}^n$ in the revised manuscript.
>
> > I think the notation around Eq. 7 could be clarified.
>
> We thank the reviewer for this observation. The reviewer's interpretations are correct. $\textbf{TP}_\ell$ denotes the projection of the standard Clebsch-Gordan tensor product to rotation order $\ell$, and $\mathbf{W}f_i$ represents per-order linear mixing, i.e. a separate learnable weight matrix applied independently to features of each rotation order. We have clarified both in the revised manuscript.
>
> > Regarding proposition 3.5 (force equilibrium), I believe this also follows directly from translational invariance of the energy.
>
> The translational invariance of energy is indeed required for force equilibrium, but it is not a sufficient condition. For example, it must also be the case that $\mathbf{\hat{F}}_{ij} = \mathbf{\hat{F}}_{ji}$ in order for the forces acting on each atom to correctly cancel.
>
> > Additional details about the ablations in Section 5.5 would be helpful.
>
> We vary the dimension of the equivariant features for the SevenNet models and both the invariant/equivariant features for the HIENet models to construct increasingly large models. We fix the number of layers and all other model design choices. In practice, we find that the specific way in which the model capacity is increased (e.g. increasing width vs. increasing depth) does not make a significant difference on the performance. Because the number of parameters is not comparable between invariant and equivariant layers, we use throughput to compare the increasingly large models. In Figure 3 and Figure 4, we report the final validation loss, which is a weighted average of the energy, force, and stress errors as defined in Appendix G.2. We choose this metric as it quantifies the models' ability to accurately predict energy, force, and stress in a single number which is easier to visualize in such a figure. We have updated Section 5.5 to clarify some of these points.
>
> [1] Complete and Efficient Graph Transformers for Crystal Material Property Prediction. Yan et al., 2024

---

> ### Comment · Reviewer_preZ · 2026-03-18
> **Most concerns resolved**
>
> Thank you - this resolves most of my concerns.
>
> One remaining issue is that I think the performance of the current state-of-the-art models should be included in the results tables in the main paper. I full understand the sentiment:
> > These contributions are orthogonal to the performance of any specific model, and we believe the hybrid design could be applied to further boost the performance of more recent architectures [...]
>
> However, comparing performance with other methods serves to give a clear picture of how the model performs relative to alternative approaches and helps place the results in context. I believe it is appropriate here to also include the best-performing models in the main results tables.

---

> > ### Author Response · Authors · 2026-03-25
> >
> > We thank the reviewer for the follow-up. We agree that contextualizing our results relative to recent models is valuable, which is why we included the comparison table and discussion of recent models on the Matbench Discovery task in the appendix. However, we prefer to keep this table in the appendix rather than moving it to the main paper. Due to computational constraints, we are unable to evaluate these models across our full benchmark suite, and we believe maintaining a consistent set of baselines throughout the paper tells a more coherent story. Moreover, the Matbench Discovery task maintains an active online leaderboard that is continuously updated, meaning any table in the main paper would quickly become outdated as new models are released. That said, to improve transparency we have expanded the discussion of newer models in the Related Work section and added a reference to Appendix E directly in the caption of Table 1.

---

### Review · Reviewer_HJNx · 2026-03-03

**Summary Of Contributions:**

This work is focused on learning interatomic potentials using HIENet, an architecture that can satisfy both invariants and equivariants. This work is very exciting and timely, and the paper is extremely well-written.

**Audience:**

Yes

**Audience Explanation:**

Machine learning has huge potential in the area of interatomic potential learning. This has the possibility of using ML to design and/or discover new materials in a trustworthy way.

**Broader Impact Concerns:**

None.

**Claims And Evidence:**

Yes

**Claims Explanation:**

The proofs show that HIENet does what it claims to do, and the results show success across the set of crystalline materials that the work targets.

**Requested Changes:**

The paper is very well-written. I'd argue that Section 4 should really be folded into Section 1, but that's my only proposed adjustment.

---

> ### Author Response · Authors · 2026-03-17
>
> We thank the reviewer for their feedback.
>
> Regarding the requested change, we agree that Section 4 is better positioned closer to Section 1, and we have updated the manuscript accordingly by moving it immediately after Section 1.

---

### Review · Reviewer_MwDg · 2026-03-04

**Summary Of Contributions:**

The paper proposes HIENet, a Machine Learning Interatomic Potential (MLIP) designed for materials discovery. The core innovation lies in its hybrid architecture, which sequences invariant message-passing layers with equivariant message-passing layers. This design aims to achieve the trade-off between the computational efficiency of purely invariant architectures and the high representational capacity of purely equivariant architectures. Furthermore, the authors strictly enforce critical physical constraints (such as force conservation, stress tensor symmetry, and O(3) equivariance) through a combination of a smooth polynomial envelope, gradient-based force/stress derivation, and specific graph construction rules. The model is evaluated on standard benchmarks (MPtrj, Matbench Discovery) and, crucially, on rigorous downstream physical tasks (phonon band structures, bulk modulus, ab initio molecular dynamics, and alloy phase diagrams).

**Key Strengths:**
1.	Comprehensive Downstream Evaluation: Unlike many MLIP papers that stop at predicting energy/forces on a validation set, this paper performs extensive real-world materials science evaluations (e.g., Phonon Band Structures, AIMD, Alloy phase diagrams).

2.	Rigorous Physical Constraints: The theoretical proofs (Section 3.5 and Appendix G) guaranteeing force conservation and O(3) equivariance are clear and robust.

3.	Strong Ablation Studies: The "Hybrid architecture generality" ablation (applying the invariant layer to SevenNet to create InvSevNet) is a brilliant addition. It convincingly demonstrates that the hybrid approach is a general design principle, not just a one-off trick.

4.	Computational Efficiency: The demonstrated speedup (90% faster than SevenNet-l3i5, 140% faster than EquiformerV2) while maintaining or exceeding accuracy is practically significant for the deployment of MLIPs.

**Key Weaknesses:**
1.	Lack of Statistical Significance: The results are presented as point estimates without error bars or standard deviations across multiple random initialization seeds.

2.	Minor Clarity Issues: The justification for O(3) vs. SO(3) graph construction is crucial to the paper's physical claims but is somewhat buried in Appendix E.

3.	Typographical Errors: There are a few noticeable typos (e.g., Table 6 caption).

**Audience:**

Yes

**Audience Explanation:**

The intersection of Machine Learning and the Physical Sciences (AI4Science) is a rapidly growing subfield within the TMLR community.
The architectural ablation showing that prepending an invariant layer to an equivariant network boosts both speed and accuracy (as seen in the InvSevNet experiment) is a valuable architectural insight for anyone designing Geometric Deep Learning models.
An MLIP that accurately predicts phonon band structures and stable AIMD trajectories while being computationally lightweight is highly desirable. The proposed methodology provides a concrete, usable tool for materials discovery.

**Broader Impact Concerns:**

No significant ethical concerns.

**Claims And Evidence:**

Yes

**Claims Explanation:**

The claims made by the authors are well-supported by both theoretical derivations and empirical results.

1.	Theoretical Claims: Propositions 3.4, 3.5, and 3.6 formally claim that the model satisfies force conservation, equilibrium, and O(3) equivariance. The mathematical proofs provided in Section 3.5 and detailed in Appendix G are sound. The use of the Gasteiger polynomial envelope to ensure smooth differentiability at the cutoff radius is a mathematically correct and elegant way to guarantee conservative forces.

2.	Empirical Claims: The claim that HIENet balances accuracy and efficiency is supported by Table 1, Table 2, and Table 5. The model outperforms the baselines on energy/force MAE while demonstrating significantly higher throughput than equivariant-only models.

3.	Physical Plausibility Claims: The claim that enforcing physical constraints leads to better downstream performance is strongly supported by the Phonon frequency evaluations (Table 3) and MD simulations (Table 6), where unconstrained models (like ORB) or models that violate energy conservation show degraded performance.

However, the empirical evidence would be fully ironclad if the authors included variance metrics.

**Requested Changes:**

**Critical Changes (Necessary for Acceptance):**
1.	Fix Typographical Errors: In the caption of Table 6, there is a glaring typo: "Table 6: MLIP !!!!prediction accuracy across...". Please thoroughly proofread the manuscript to fix this and any other grammatical/formatting issues.

2.	Address Variance/Reproducibility: Please clarify whether the results reported in Tables 1 and 2 are from a single run or averaged over multiple seeds. Given the computational cost (250 epochs on 8xA100s), if only a single run was feasible, this must be explicitly stated in the limitations section. If multiple runs are feasible, please provide standard deviations to ensure the performance gaps (e.g., against EquiformerV2) are statistically significant.

3.	Clarify O(3) Graph Construction: Please move a brief summary of the ablation currently in Appendix E (Table 8) into the main text (around Section 3.2) to better justify this design choice.

**Strengthening Changes (Highly Recommended):**
1.	Discussion on Layer Ordering: Table 9 shows that the "Invariant First" ordering performs best. It would strengthen the paper to provide a deeper intuitive or theoretical hypothesis in the main text as to why building invariant representations first, before upgrading to equivariant tensor products, yields a more optimal learning trajectory than "Equivariant First" or alternating layers.

---

> ### Author Response · Authors · 2026-03-17
>
> We thank the reviewer for their feedback and provide response to all questions/points raised.
>
> > Fix Typographical Errors: In the caption of Table 6, there is a glaring typo: "Table 6: MLIP !!!!prediction accuracy across...".
>
> We thank the reviewer for catching this. We have fixed this typo and performed a thorough proofread of the manuscript to fix several additional grammatical/formatting issues.
>
> > Address Variance/Reproducibility: Please clarify whether the results reported in Tables 1 and 2 are from a single run or averaged over multiple seeds.
>
> The results in Tables 1 and 2 are from a single training run. Given the substantial computational cost of training HIENet, running multiple seeds is unfortunately not feasible within our current computational budget. We note that reporting standard deviations is not standard practice in this research area, as none of the baseline models we compare against provide such statistics either. We have added an explicit statement about this in the limitations section of the revised manuscript.
>
> > Clarify O(3) Graph Construction: Please move a brief summary of the ablation currently in Appendix E into the main text to better justify this design choice.
>
> We have moved this ablation into the main text in Section 5.6, which is referenced directly from the O(3) graph construction discussion in the updated Section 4.1. We have also strengthened the surrounding discussion by explicitly connecting the symmetry mismatch introduced by SO(3) encodings to the observed performance drop.
>
> > Discussion on Layer Ordering: Table 9 shows that the "Invariant First" ordering performs best. It would strengthen the paper to provide a deeper intuitive or theoretical hypothesis in the main text.
>
> We have added a discussion of this in Section 5.6. Our intuition is that the invariant layer first aggregates local chemical context including coordination patterns and species identities without the computational overhead of tensor products, giving the equivariant layers better scalar features to lift into higher-order geometric features. Without this, tensor products must simultaneously learn both chemical and geometric structure, leading to the degraded performance observed in the equivariant-first and alternating configurations.

---

> > ### Comment · Reviewer_MwDg · 2026-03-19
> >
> > Thank you for your clarifications. I have no further concerns.

---

### Decision · Action_Editor_fKnt · 2026-04-24

**Recommendation:** Accept as is

**Audience:**

Yes

**Audience Explanation:**

Achieving both accuracy and efficiency in a force field has always been longed for in molecular science, where the accuracy is for deriving stable and realistic simulations and efficiency is for repeated calling of the force field in simulation. The work proposes a design that seems to have achieved a better trade-off than using invariant model and equivariant model alone. Demonstration of MD, the real use case, is also provided.

**Claims And Evidence:**

Yes

**Claims Explanation:**

The paper presented an MLIP architecture aiming both accuracy and efficiency. The general idea is to only keep the invariant feature in the forward data stream and treat distance feature and tensorial features as augmented information for updating the invariant feature, so that invariant layers and tensor-product-based equivariant layers can be concatenated. The hope is that leveraging the invariant layers could release computational cost while keeping the level of accuracy of a purely equivariant model. This could be a reasonable approach, and experimental results provide a comprehensive support, including better efficiency and accuracy in both single-point prediction and more valuably, in MD statistics. Comprehensive ablation studies are conducted supporting the desired effect of the design. Conservativeness is guaranteed and formally stated by the taking derivatives of the scalar energy (an invariant feature) together with smoothed cutoff function. The reviewers mentioned a few presentation issues, which seem to have been properly addressed.